# OMAC: A Broad Optimization Framework for LLM-Based Multi-Agent Collaboration

## Abstract

Agents powered by advanced large language models (LLMs) have demonstrated impressive capabilities across diverse complex applications. Recently, Multi-Agent Systems (MAS), wherein multiple agents collaborate and communicate with each other, have exhibited enhanced capabilities in complex tasks, such as high-quality code generation and arithmetic reasoning. However, the development of such systems often relies on handcrafted methods, and the literature on systematic design and optimization of LLM-based MAS remains limited. In this work, we introduce **OMAC**, a general framework designed for holistic optimization of LLM-based MAS. Specifically, we identify five key optimization dimensions for MAS, encompassing both agent functionality and collaboration structure. Building upon these dimensions, we first propose a general algorithm, utilizing two actors termed the Semantic Initializer and the Contrastive Comparator, to optimize any single dimension. Then, we present an algorithm for joint optimization across multiple dimensions. Extensive experiments demonstrate the superior performance of OMAC on diverse tasks against recent approaches. Code and data are available at: https://anonymous.4open.science/r/OMAC-Sub-3FF8.

## 1 Introduction

Autonomous agents leveraging advanced large language models (LLMs) have shown significant potential in addressing complex problems and executing diverse tasks Shinn et al. (2023). Recently, employing multiple collaborating agents has emerged as a prominent research direction for overcoming the inherent limitations of single-agent approaches in handling tasks within sophisticated environments (Li et al. (2023a); Wang et al. (2023b)). An advanced collaborative paradigm also involves a multi-step process wherein agents sequentially resolve tasks by utilizing outputs from preceding steps. This Multi-Agent Systems (MAS) approach has already yielded substantial advancements in various applications, such as code generation Shinn et al. (2023), reasoning Du et al. (2023), and decision making Sun et al. (2024).

However, the design of existing MAS predominantly relies on hand-crafted strategies. Regarding agent construction, prior studies typically employ methods based either on human expertise Liu et al. (2023) or LLM generation Wang et al. (2023b). Although some research has explored optimizing the functionality of a single agent via techniques such as fine-tuning Huang et al. (2024) or prompt-tuning Wang et al. (2023a), these efforts don't explicitly address agent optimization within the context of MAS involving multi-step collaborative processes. Concerning collaboration structures, existing approaches typically define fixed architectures tailored to specific application scenarios, encompassing cooperative tasks (e.g., code generation Dong et al. (2023), decision-making Qian et al. (2023)) and competitive tasks (e.g., debate Du et al. (2023)). However, the design of these structures predominantly relies on human prior knowledge or particular empirical findings, thereby limiting their generality and the flexibility required for autonomous optimization towards more effective structural configurations. A recent study, DyLAN Liu et al. (2024), represents a promising effort toward optimizing collaboration structures in MAS. However, its approach centers solely on the unsupervised optimization of agent team composition, employing metrics derived from preliminary trials and LLM-based judgments, rather than utilizing supervised signals for validation and optimization.

To address these challenges, we introduce **OMAC**, a comprehensive framework designed for the integrated optimization of MAS. Specifically, we identify and summarize **five key optimization**

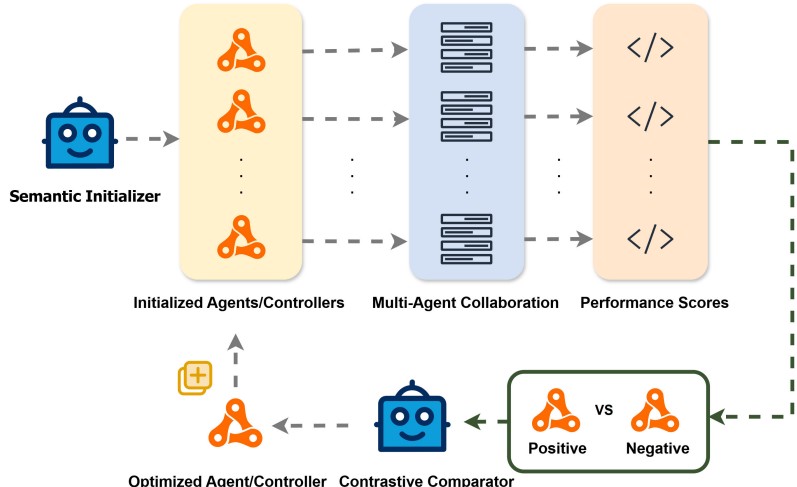

Figure 1: OMAC optimization workflow for a single dimension.

**dimensions** pertaining to both agent functionality and collaboration structure. The functional dimensions include optimizing existing agents and constructing new agents tailored for collaboration. Regarding the structural aspect, we optimize[1] LLM-based controllers to manage collaboration structures, encompassing decisions on the overall candidate agent teams, dynamic agent selection for individual collaboration steps, and the mechanisms governing inter-agent communication.

Then we first propose a general algorithm employing two LLM-powered actors, the **Semantic Initializer** and the **Contrastive Comparator**, to optimize any individual dimension. Specifically, the Semantic Initializer leverages the knowledge and reasoning capabilities of LLMs to generate an initial collection of agents or controllers corresponding to the dimension being optimized by exploring the relevant semantic space. Subsequently, each initialized agent or controller undergoes evaluation within the multi-agent collaboration process on the training set to determine its performance score. A positive-negative pair (high-performing and low-performing) is then sampled based on these scores. The Contrastive Comparator then performs contrastive reasoning on this pair to identify factors underlying the performance gap, subsequently generating a refined agent or controller. This refined agent/controller is then re-evaluated via the collaboration process, and the cycle of sampling, contrastive comparison, and refinement is iterated. Beyond single-dimension optimization, we further propose an algorithm for iterative, joint optimization across multiple dimensions, enabling further MAS enhancement through synergistic optimization of agents and controllers. More implementation details of the actors and optimization algorithm are illustrated in Section 3.2.

We conduct extensive experiments across diverse tasks, including general reasoning, code generation, and arithmetic reasoning. The results demonstrate that OMAC consistently outperforms strong baselines when optimizing each of the five individual dimensions. Furthermore, joint optimization across multiple dimensions is shown to further significantly enhance MAS's performance.

Our key contributions can be summarized as follows:

- We introduce OMAC, a general supervised framework for optimizing multi-agent systems engaged in multi-step collaboration. To achieve this, we identify five key optimization dimensions covering both agent functionality and collaboration structure.

- We propose a general algorithm, utilizing two actors termed the Semantic Initializer and the Contrastive Comparator, for optimizing each of the five dimensions. Furthermore, we present an algorithm enabling iterative, joint optimization across multiple dimensions.

- We evaluate OMAC on benchmark tasks including general reasoning, code generation, and arithmetic reasoning. Empirical results demonstrate that OMAC significantly outperforms existing approaches through the automated optimization of functional and structural MAS designs.

---

[1]Note that we use the word "optimize" in a colloquial way to signify improvement instead of mathematical guarantee of an optimal solution.

## 2 PROBLEM DEFINITION

We study the optimization of MAS within the context of multi-step collaboration process. The definition of agents and the fundamental collaboration workflow are as follows:

**Definition 1: Agents.** An LLM-powered agent is governed by natural language prompts to generate solutions for completing a given task. Formally, we define an agent $a$ as a function: $\mathcal{A} : \mathcal{P} \times \mathcal{I} \to \mathcal{O}$. Here, $p \in \mathcal{P}$ represents the instruction prompt defining the role and functionality of the agent. In an in-context learning setting, the prompt may also include few-shot examples serving as demonstrations to guide the agent's behavior. The input $i \in \mathcal{I}$ typically comprises the query information, task description, and potentially instructions or solutions from other agents in MAS. The output $o \in \mathcal{O}$ generally encompasses the agent's generated analyses, reasoning, and solutions to the given task.

**Definition 2: Multi-Step Collaboration.** In this work, we consider multiple agents collaborating within a multi-step procedure. Typically, each step $s_t \in \mathcal{S}$ involves a subset of agents from the overall team, denoted as $\{a_{t,1}, a_{t,2}, ..., a_{t,n}\} \subseteq s_t$. This multi-step approach aims to enhance problem-solving by enabling agents at each step to leverage solutions and outcomes produced by agents in preceding steps. For instance, when solving a complex mathematical problem, agents in the initial step may first decompose the problem into a series of simpler subproblems. Agents in subsequent steps can then address each subproblem by integrating analyses and solutions generated earlier with relevant contextual information. Especially, we propose determining and refining this collaboration structure (e.g., selecting agents to participate at each step) using **LLM-based controllers**. The functionalities of these controllers are detailed in Section 3.1.

## 3 METHODOLOGY

In this section, we first detail the five dimensions we identified for optimizing multi-agent collaboration, covering both agent functionality and collaboration structure. Subsequently, we present our proposed algorithm for optimizing each dimension individually. Finally, we describe our algorithm for iteratively and jointly optimizing multiple dimensions.

### 3.1 FIVE DIMENSIONS FOR MULTI-AGENT COLLABORATION OPTIMIZATION

As discussed in Section 2, agent functionality and collaboration structure are fundamental components of MAS. For the holistic optimization of such systems, we identify five key dimensions: two related to agent functionality and three concerning the collaboration structure, as detailed below:

- **Optimizing existing agents (Fun-1).** This dimension focuses on refining an existing agent within the MAS. Specifically, the goal is to optimize the agent's instruction prompt and/or its associated few-shot examples (in few-shot learning settings) to enhance its task-specific performance.

- **Optimizing construction of new agents (Fun-2).** This dimension addresses the creation of new agents. Specifically, given the task context and existing MAS configuration, the objective is to generate and optimize the instruction prompt and/or few-shot examples used to power a new LLM-based agent. This newly constructed agent will then be integrated into the existing MAS, enhancing the overall collaborative capability in completing the given task.

- **Optimizing candidate agent selection (Str-1).** This dimension involves selecting suitable candidate agents from all available agents given the specific task before the collaboration. Specifically, the objective is to optimize the instruction prompt of an LLM-based controller. This prompt directs the controller to identify the most beneficial subset of agents for the multi-agent collaboration process, a decision informed by the provided task context and functionalities of existing agents.

- **Optimizing dynamic agent participation (Str-2).** This dimension concerns the dynamic selection of agents for participation at individual steps of the multi-step collaboration. Specifically, the objective is to optimize the instruction prompt of an LLM-based controller to choose the most suitable agents from the candidate team for participation in the current collaboration step. In contrast to Str-1, this controller additionally incorporates output solutions from previous agents as contextual input, enabling the dynamic selection of agents anticipated to contribute most effectively and efficiently at the current step of multi-agent collaboration.

- **Optimizing agent communication flows (Str-3).** This dimension addresses the optimization of communication flows among agents. Specifically, the goal is to optimize the instruction prompt of an LLM-based controller to determine whether the output from one agent should serve as input context for another agent during collaboration. The controller, guided by the optimized prompt, makes these communication routing decisions based on the given task context and the involved agents' functionalities.

These five proposed dimensions are designed to comprehensively cover the fundamental optimizable aspects of multi-step multi-agent collaboration. We derive them by conceptualizing the collaboration process in MAS as information flow over a graph, where agents correspond to nodes and communication channels to edges. Thus, the two functional dimensions optimize node capabilities, either improving existing agents or constructing new ones. And the three structural dimensions define graph construction, encompassing global and dynamic local agent selection, as well as inter-agent communication routes. Together, these five dimensions comprehensively capture the essential aspects required to optimize the information-flow graph of a multi-agent system. More illustrations of the underlying intuition, implementation details, and examples are given in Appendix A and Appendix D.2.

### 3.2 Optimization for a Single Dimension

We first propose a unified algorithm designed for optimizing a single dimension. This algorithm employs two core LLM-powered actors: the Semantic Initializer and the Contrastive Comparator. Notably, our algorithm is generally applicable to any of the five dimensions identified before, requiring only minor adaptations to the contextual information supplied to the Semantic Initializer and Contrastive Comparator. The overall framework is shown in Figure 1.

**Initialization of Collection.** The first step involves generating an initial collection of agents or controllers corresponding to the dimension being optimized. Specifically, we utilize an LLM-powered actor named the Semantic Initializer to accomplish this. For each of the five dimensions, we instruct the Semantic Initializer by first describing essential contextual information regarding the existing MAS configuration and the given task. Next, we specify the expected functionality of the generated prompts according to the dimension being optimized: either to power an agent (Fun-1 and Fun-2) or to guide the controller managing the collaboration structure (Str-1 to Str-3). Then, we provide a one-shot example and then indicate the number of initial agents/controllers (i.e., their instruction prompts) to be generated. In addition to generating instruction prompts, if the optimized agents or controllers operate under a few-shot learning setting, the Semantic Initializer can also be instructed to generate few-shot examples using the same procedure.

The rationale behind this design is to leverage the knowledge and reasoning capabilities of LLMs to systematically explore the semantic space for instructing an agent or controller. This exploration adheres to specified functionality corresponding to the dimension being optimized, while simultaneously introducing controlled variations in focal emphasis and implementation details. Such variations promote diversity and inject stochasticity into the initial collection, which is utilized by subsequent optimization with contrastive reasoning.

**Evaluation and Sampling.** After obtaining the initial collection of agents or controllers to be optimized, we first evaluate their performance by integrating each one into the MAS with other existing agents/controllers and executing the collaboration process on the training data. Performance scores are then calculated based on the MAS's overall performance associated with each agent/controller in the initial collection. Subsequently, we sample a positive-negative pair of agents/controllers based on these performance scores. Specifically, we define two thresholds, $h$ and $l$, as upper and lower bounds for selecting the positive and negative ones. Given the current collection of size $n$, the top $\lfloor n \times h \rfloor$ performing ones are classified as positive, whereas the bottom $\lfloor n \times l \rfloor$ are classified as negative. A positive-negative pair is then randomly sampled from these two groups. The rationale behind sampling within thresholds is to diversify the positive-negative pairs obtained in each iteration, thereby enhancing the generality and robustness of subsequent contrastive reasoning.

**Contrastive Reasoning for Optimization.** Once we get the positive-negative pair, we feed it into the Contrastive Comparator. This LLM-powered comparator is tasked to carefully compare this pair of agents/controllers and reason the underlying factors for their performance gap. Then, it is instructed to generate a new agent/controller (i.e., its instruction prompt and/or the few-shot examples) that is

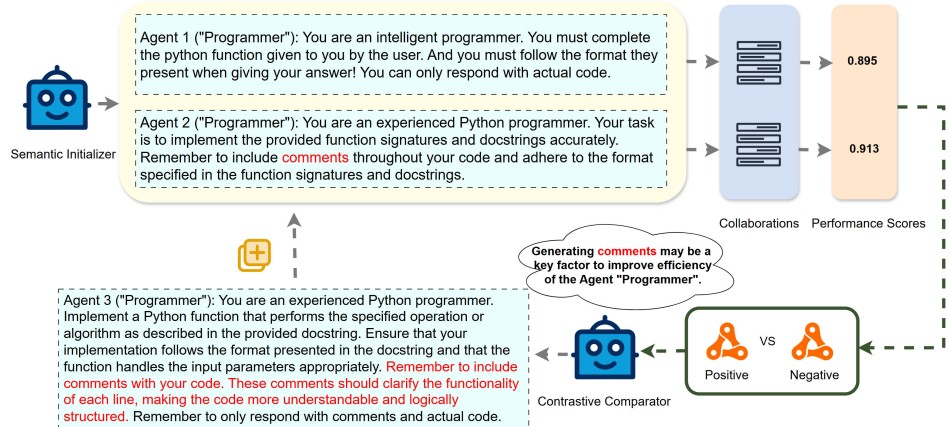

Figure 2: An example of optimization of a single dimension of OMAC.

expected to perform even better than the positive example. Similarly, information regarding the given task and the dimension being optimized is provided as the context for the reasoning. After that, this newly generated agent/controller is incorporated into the initial collection and evaluated through the collaboration process. The cycle of evaluation, sampling, contrastive comparison and refinement is then repeated until reaching the predefined maximum number of iterations.

The rationale here is to leverage the advanced reasoning capabilities of LLMs to analyze the performance difference by contrasting the positive-negative pair with the given task context. This performance gap constitutes a supervised signal derived from evaluations on the training data. By reasoning about this gap, the LLM can refine the corresponding instruction prompts, aiming to enhance or amplify factors correlated with positive performance while mitigating or removing factors associated with negative performance.

**Demonstration Example.** Figure 2 illustrates an example of optimizing an existing agent functioning as a "Programmer" (Fun-1) using OMAC. The Semantic Initializer first generates two prompts, both designed to instruct an agent with the role as a Programmer for the code generation task. The primary difference between the prompts is that the second one explicitly requires including comments throughout the code. Each prompt is then evaluated by the overall performance of the MAS on the training data. Subsequently, a positive-negative pair is sampled and provided to the Contrastive Comparator. By analyzing the performance difference, the Contrastive Comparator identifies code commenting as a key factor enhancing the programmer agent's performance. Consequently, it generates a new prompt emphasizing the importance of comments and elaborating on their purpose and proper usage. Ideally, this refined prompt can guide the programmer agent to produce more accurate and logical code, further improving the overall performance of MAS.

### 3.3 OPTIMIZATION FOR MULTIPLE DIMENSIONS

Beyond the single dimension optimization, we further propose an algorithm for jointly optimizing multiple dimensions. Specifically, our method iteratively optimizes each dimension individually while keeping the other dimensions fixed. For example, to jointly optimize an existing agent (Fun-1) and the controller to select candidate agents (Str-1), we first execute the single dimension optimization described in Section 3.2 for Fun-1. After that, we retain the agent from the agent collection exhibiting the highest performance score and subsequently optimize the controller responsible for candidate selection (Str-1). After deriving the optimized controller, we repeat this iterative process until predefined termination conditions (e.g., reaching a maximum number of iterations) are satisfied. Figure 3 illustrates this process.

The motivation of iterative optimization is to maintain the effectiveness of contrastive reasoning. Specifically, by limiting variations within positive-negative pairs to a single dimension at a time, we ensure consistency across other factors. Consequently, the Contrastive Comparator can clearly identify the reasons for performance differences, thus avoiding complexities arising from multiple interacting variables simultaneously affecting overall performance.

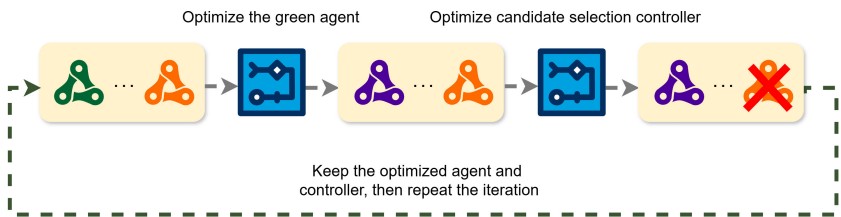

Optimize the green agent          Optimize candidate selection controller

Keep the optimized agent and
controller, then repeat the iteration

Figure 3: OMAC optimization framework for multiple dimensions.

### 3.4 INFERENCE AND COMPUTATION EFFICIENCY

**Inference.** After optimizing with either single dimension or multiple dimensions, we select the agent(s) and/or controller(s) configuration that yielded the highest performance on the training set. These optimized agents/controllers are then utilized within the MAS to conduct inference and evaluation on the test set.

**Computational Efficiency.** In Section 3.3, we introduced our algorithm for iteratively optimizing multiple dimensions jointly. However, iterative optimization may result in substantial computational demands due to the exponential growth of dimension combinations. For instance, mutually optimizing two dimensions over two iterations results in four times the computational cost compared to optimizing a single dimension. To mitigate this, we propose selectively incorporating only those dimensions that individually demonstrate the most significant performance improvements into the iterative joint optimization process. We empirically validate the effectiveness of this strategy in our experiments.

## 4 EXPERIMENTS

**Tasks and Datasets.** We primarily evaluate OMAC across three task domains: code generation, general reasoning, and arithmetic reasoning. For code generation, we utilize the HumanEval benchmark Chen et al. (2021), which contains human-authored function-level code completions accompanied by unit tests. We employ Pass@1 as the evaluation metric, representing the proportion of generated code solutions that successfully pass the unit tests. For general reasoning tasks, we leverage the MMLU dataset Hendrycks et al. (2021a), which comprises multiple-choice questions across humanities, social sciences, hard sciences, and other fields. We measure performance using answer accuracy. For arithmetic reasoning, we use the MATH dataset Hendrycks et al. (2021b), encompassing mathematical problems spanning seven subareas, again employing accuracy as our evaluation metric. Additional details on tasks and datasets can be found in Appendix B.1. Besides these classic benchmarks, we further conducted experiments on another more complex benchmarks, MBPP Austin et al. (2021) and GAIA Mialon et al. (2023). Due to the space limitation, we report relevant results on MBPP and GAIA in Appendix B.2.5.

**Baselines.** As prior studies typically adopt different agent functionalities and collaboration structures tailored to specific applications, we incorporate distinct baselines for each task domain. Specifically, we compare against single-agent methods, multi-agent methods with fixed configurations, and DyLAN Liu et al. (2024), ADAS Hu et al. (2025), and AFlow Zhang et al. (2025c) for MAS optimization.

For code generation tasks, we select CodeT Chen et al. (2023) as the single-agent baseline and AgentVerse Chen et al. (2024a) as the multi-agent approach. For general reasoning and arithmetic reasoning tasks, we utilize a single agent directly generating answers as the single-agent baseline, denoted as Single Execution (SE). For multi-agent baseline, we compare LLM Debate Du et al. (2023). All baseline approaches retain their original configurations to ensure fair comparisons.

**OMAC Setup.** OMAC is designed to optimize the functionality and collaboration structure of an existing MAS. As such, we adopt the agent designs and collaboration structures from the SOTA method DyLAN Liu et al. (2024) as the default configuration for OMAC across all datasets. Specifically, the default MAS includes 7 agents for code generation, 7 agents for general reasoning, and 4 agents for arithmetic reasoning tasks. Names and functionalities of these agents are detailed in Appendix B.1. The default collaboration structure is "fully-connected", meaning all existing agents participate

Table 1: Performance on code generation tasks with single-dimension optimization.

| Method | Baselines | | | | | OMAC (Structural Dimension) | | |
|---|---|---|---|---|---|---|---|---|
| | CodeT | AgentVerse | ADAS | AFlow | DyLAN | Str-1 | Str-2 | Str-3 |
| Pass@1 | $67.50_{\pm1.68}$ | $78.29_{\pm2.34}$ | $75.61_{\pm2.06}$ | $85.63_{\pm2.10}$ | $\underline{85.74}_{\pm2.83}$ | $\mathbf{86.76}_{\pm1.22}$ | $\mathbf{86.92}_{\pm2.27}$ | $\mathbf{87.55}_{\pm2.46}$ |

| Method | OMAC (Functional Dimension) | | | | | | | |
|---|---|---|---|---|---|---|---|---|
| | Fun-1.1 | Fun-1.2 | Fun-1.3 | Fun-1.4 | Fun-1.5 | Fun-1.6 | Fun-1.7 | Fun-2 |
| Pass@1 | $\mathbf{88.39}_{\pm2.54}$ | $\mathbf{86.31}_{\pm2.21}$ | $\mathbf{88.87}_{\pm1.36}$ | $\mathbf{89.25}_{\pm1.30}$ | $\mathbf{88.74}_{\pm2.67}$ | $\mathbf{88.39}_{\pm1.22}$ | $\mathbf{88.34}_{\pm1.42}$ | $\mathbf{86.77}_{\pm2.43}$ |

Table 2: Performance on general reasoning tasks with single-dimension optimization.

| Method | Baselines | | | | | OMAC (Structural Dimension) | | |
|---|---|---|---|---|---|---|---|---|
| | SE | LLM Debate | ADAS | AFlow | DyLAN | Str-1 | Str-2 | Str-3 |
| Accuracy | $65.76_{\pm2.31}$ | $68.74_{\pm2.67}$ | $69.02_{\pm2.44}$ | $\underline{70.06}_{\pm2.57}$ | $69.42_{\pm2.16}$ | $\mathbf{73.14}_{\pm2.24}$ | $\mathbf{72.06}_{\pm1.96}$ | $\mathbf{73.18}_{\pm1.47}$ |

| Method | OMAC (Functional Dimension) | | | | | | | |
|---|---|---|---|---|---|---|---|---|
| | Fun-1.1 | Fun-1.2 | Fun-1.3 | Fun-1.4 | Fun-1.5 | Fun-1.6 | Fun-1.7 | Fun-2 |
| Accuracy | $\mathbf{72.33}_{\pm2.47}$ | $\mathbf{73.23}_{\pm1.83}$ | $\mathbf{72.06}_{\pm2.41}$ | $\mathbf{74.22}_{\pm2.22}$ | $\mathbf{73.15}_{\pm2.86}$ | $\mathbf{72.07}_{\pm2.92}$ | $\mathbf{71.83}_{\pm2.74}$ | $\mathbf{71.02}_{\pm2.57}$ |

in each step, and each agent in the current step receives all outputs from agents in the previous step as input. We utilize the same evaluation metrics on each dataset described above to construct positive-negative pairs, setting thresholds $l = h = 0.5$ consistently. Regarding other hyperparameters, the Semantic Initializer generates an initial collection of size 3, and we set the maximum number of contrastive reasoning iterations to 3. For fair comparisons, we employ `gpt-3.5-turbo-1106` with temperature of 0.8 as the base LLM across all baselines and our OMAC. Experiments with other base LLM are reported in Appendix B.2.4. All experiments are repeated three times, and the mean and standard deviation are reported. More implementation details are provided in Appendix B.1.

## 4.1 SINGLE-DIMENSION OPTIMIZATION RESULTS

We first employ OMAC to individually optimize each of the five dimensions defined in Section 3.1. Specifically, for Fun-1, we optimize each existing agent in the default MAS separately, leading to sub-dimensions denoted as Fun-1.1 to Fun-1.7 for 7 agents in general reasoning and code generation tasks. For arithmetic reasoning tasks, we follow DyLAN to employ four agents sharing the same prompts and few-shot examples; hence, we optimize either the instruction prompts or few-shot examples for all of them, resulting in sub-dimensions Fun-1.1 and Fun-1.2. Table 2, Table 1, and Table 3 present results on each benchmark respectively.

The results indicate that optimizing each of the five dimensions individually leads to significant performance improvements across all three tasks in most cases. Note that the average relative optimization gains achieved by OMAC (3.6%, 2.8%, and 4.9% across the three benchmarks per dimension) are significant and comparable to those reported by recent MAS optimization studies, such as DyLAN Liu et al. (2024) under similar experimental conditions. These results validate our delineation of the five optimization dimensions for multi-agent collaboration and confirm the effectiveness of our single-dimension optimization algorithm. Furthermore, comparisons between single-agent and multi-agent methods highlight the advantages of leveraging multiple agents for complex tasks. Finally, while both DyLAN and OMAC achieve improvements through structural optimization, our approach demonstrates superior efficacy by utilizing supervised signals derived from training data evaluations. More experiments and results examining the effects of hyper-parameters, such as the size of the initial collection, the maximum number of iterations, and the sampling thresholds, are presented in Appendix B.2.1.

Table 3: Performance on arithmetic reasoning tasks with single-dimension optimization.

| Method | Baselines | | | | |
|---|---|---|---|---|---|
| | SE | LLM Debate | ADAS | AFlow | DyLAN |
| Accuracy(%) | $28.72_{\pm1.75}$ | $29.42_{\pm2.33}$ | $28.94_{\pm2.04}$ | $32.49_{\pm2.62}$ | $32.35_{\pm1.94}$ |

| Method | OMAC (Structural Dimension) | | | OMAC (Functional Dimension) | | |
|---|---|---|---|---|---|---|
| | Str-1 | Str-2 | Str-3 | Fun-1.1 | Fun-1.2 | Fun-2 |
| Accuracy(%) | $\mathbf{33.34}_{\pm1.45}$ | $\mathbf{33.41}_{\pm1.37}$ | $\mathbf{33.70}_{\pm1.62}$ | $\mathbf{35.17}_{\pm1.96}$ | $\mathbf{34.91}_{\pm2.03}$ | $\mathbf{33.95}_{\pm1.30}$ |

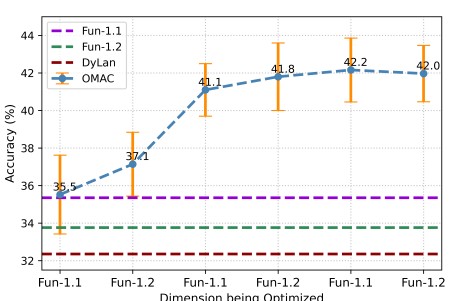 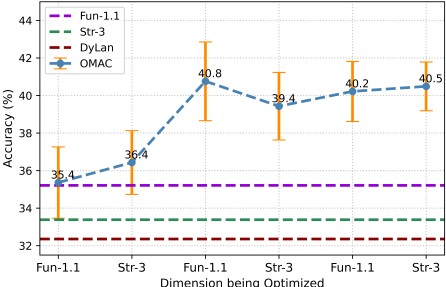

Figure 4: Performance during iterative optimization for multiple dimensions on arithmetic reasoning tasks. The X-axis represents the dimensions undergoing three iterations of optimization, while each point indicates the MAS's performance with the optimized dimensions on the test set. The error bar denotes the standard deviation.

## 4.2 MULTI-DIMENSION OPTIMIZATION RESULTS

We further experiment on iteratively optimizing multiple dimensions with OMAC. Following the strategy outlined in Section 3.4 for computational efficiency, we selectively incorporate only the dimensions exhibiting the most substantial performance improvements into our multi-dimensional optimization process. Specifically, we experiment by jointly optimizing the two best-performing dimensions by individual optimization (which, across all tasks, correspond to the two functional dimensions), as well as pairing the best functional dimension with the best structural dimension. These selected dimensions are then iteratively optimized following the procedure illustrated in Section 3.3, repeating for three iterations.

Figure 4 presents the experimental results on arithmetic reasoning tasks. Similar trends were observed in code generation and general reasoning tasks, where the results are provided in Appendix B.2.3. The figures clearly illustrate that iterative optimization of multiple dimensions **yields substantially larger performance improvements compared to optimizing only a single dimension** (e.g, improve from 2.9% to 9.6% in Figure 4). Furthermore, jointly optimizing dimensions that individually demonstrate the most significant improvements consistently yields greater benefits compared to optimizing suboptimal dimension pairs. This supports the effectiveness of our dimension-selection strategy for mitigating the computational overhead with multi-dimensional optimization. Additional experiments validating the iterative optimization design are reported in Appendix B.2.3.

## 4.3 ABLATION STUDY

We further conduct experiments to validate the two actors of our optimization algorithm: the Semantic Initializer and the Contrastive Comparator. Specifically, we introduce an ablation model, OMAC-C, which excludes the Contrastive Comparator from the optimization pipeline described in Section 3.2. Thus, OMAC-C comprises of only the Semantic Initializer, which generates an initial collection of agents or controllers. Each generated agent/controller is subsequently evaluated through the MAS collaboration process on the training set, after which the agent or controller achieving the highest performance score is selected for evaluation on the test set.

Table 4: Accuracy (%) of OMAC-C and OMAC on arithmetic reasoning task.

| Method | Str-1 | Str-2 | Str-3 | Fun-1.1 | Fun-1.2 | Fun-2 |
|---|---|---|---|---|---|---|
| OMAC-C | $32.64_{\pm 1.98}$ | $32.67_{\pm 2.10}$ | $32.76_{\pm 2.31}$ | $34.20_{\pm 2.87}$ | $33.69_{\pm 2.32}$ | $32.71_{\pm 2.03}$ |
| OMAC | $33.34_{\pm 1.45}$ | $33.41_{\pm 1.37}$ | $33.70_{\pm 1.62}$ | $35.17_{\pm 1.96}$ | $34.91_{\pm 2.03}$ | $33.95_{\pm 1.30}$ |

Table 4 presents the results on arithmetic reasoning tasks. The comparison between OMAC-C and OMAC clearly demonstrates the significant advantage provided by the Contrastive Comparator, which leverages contrastive reasoning to further optimize the agents or controllers generated by the Semantic Initializer. Nevertheless, OMAC-C consistently outperforms the strongest baseline, DyLAN, across all optimization dimensions. It indicates that even solely exploring the semantic space via LLM-based initialization for the agents or controllers in MAS contributes meaningful performance improvements. Additional results for the other two tasks are provided in Appendix B.2.2.

## 4.4 COMPUTATION COST

In general, OMAC reduces inference-time computational cost relative to baselines, owing to its dynamic agent selection and communication flow optimization. During training, we further investigated the trade-off between training data size and computational cost, leading to valuable insights and findings. For completeness, we present the detailed results and discussions in Appendix B.2.6.

## 5 RELATED WORK

**Multi-Agent Systems.** Multi-Agent Systems (MAS) that leverage multiple LLM-based agents communicating and collaborating, are increasingly employed to tackle complex, multi-step challenges Tran et al. (2025). Existing research and industrial applications have demonstrated the effectiveness and robustness of these systems across diverse domains, such as code generation (Li et al. (2023a); Barbarroxa et al. (2024)), reasoning Du et al. (2023), question answering Das et al. (2023), and decision making (Nascimento et al. (2023); Sun et al. (2024)). Two critical factors in designing MAS are agent composition and the collaboration structure Tran et al. (2025). We categorize these factors as relating to the functional and structural properties of MAS, respectively.

**Agent Construction in MAS.** The construction of agents directly determines the system's overall functional capabilities. Most existing works on MAS employ either manually designed prompts or prompts generated by LLMs to construct agents tailored to specific application scenarios. For example, Das et al. (2023) constructed agent teams with specific roles using hand-crafted instruction prompts for software development. Wang et al. (2023b) utilized LLMs to generate role-specific prompts for agents in response to task-specific queries. However, both manual prompt design and LLM-driven agent generation rely heavily on human prior knowledge, requiring empirical verification through trial-and-error. Although previous studies have explored agent optimization through fine-tuning Li et al. (2023b) or prompt-tuning techniques Khattab et al. (2023), these methods do not explicitly address the optimization of agents within MAS involving multi-step collaborative processes.

**Collaboration Structure in MAS.** Another crucial aspect of MAS is the collaboration structure, which defines how candidate agents collaborate and communicate to ultimately resolve the given task. Existing research has proposed various structures, such as centralized Jiang et al. (2023), decentralized Liang et al. (2023), and hierarchical Hao et al. (2023) approaches. However, these structures are typically manually designed for specific task categories and remain static throughout the collaboration process. A recent work, DyLAN Liu et al. (2024), represents an initial effort in dynamically optimizing the collaboration structure within MAS. However, its focus is limited to optimizing agent team composition, relying on an unsupervised metric called "Agent Importance Score" computed using heuristic rules combined with LLM-based judgments. In contrast, OMAC comprehensively examines both functional and structural optimization of MAS in a joint supervised manner, addressing more fine-grained optimization of collaboration structures.

**Gradient-Based Agentic Optimization Methods.** As previously discussed, prior research has explored several gradient-based optimization methods for agent systems, such as supervised fine-tuning Li et al. (2023b) and prompt-tuning techniques Khattab et al. (2023). However, most of these

approaches are designed for single-agent optimization and do not extend naturally to MAS that require multi-step collaborative interactions among multiple agents. To the best of our knowledge, research on gradient-based methods specifically targeting MAS optimization in multi-step collaboration settings remains very limited.

In scenarios involving multiple interacting agents, gradient-based optimization becomes particularly challenging due to the presence of numerous confounding variables and the difficulty of jointly performance optimization with multiple functional and structural components. For instance, while some works have explored the use of reinforcement learning for optimizing agentic systems, these methods typically focus on single-agent settings (Shao et al. (2024); Qian et al. (2025)), which differ fundamentally from the multi-agent optimization problem addressed in our study. A few very recent works, such as Spiral Liu et al. (2025) and MHGPO Chen et al. (2025), have made initial attempts at applying RL to MAS optimization. However, their formulations are generally constrained to single-interaction and shared-policy scenarios, where all agents share the same control policy and perform only one round of communication or problem-solving. This setup is considerably simpler and less flexible than the setting studied in our work, which involves multi-step collaboration among role-specialized agents operating within dynamic interaction structures. Consequently, these approaches are not directly comparable to OMAC, which provides a general and extensible optimization framework for optimizing multi-step, role-specialized collaborations among multiple agents.

**Non-Gradient-Based MAS Optimization Methods.** A very recent work, ADAS Hu et al. (2025), takes an initial step toward applying prompt evolution techniques combined with search tools to optimize MAS. However, ADAS focuses on agent functionality optimization, leaving the collaboration structure predefined and fixed. This contrasts with OMAC, which introduces a general framework capable of comprehensively optimizing both the functional and structural aspects of MAS. Beyond ADAS, a couple of recent studies have explored non-gradient-based optimization for MAS from other perspectives. For example, AFlow Zhang et al. (2025c) leverages supervised signals to guide Monte Carlo Tree Search (MCTS) for discovering effective agentic workflows. In contrast, OMAC uses supervised signals directly for contrastive reasoning–based optimization, representing a distinct paradigm that avoids reliance on search-based methods. Similarly, other recent works such as G-Designer Zhang et al. (2025b) and MaAS Zhang et al. (2025a) focus on architectural search for identifying effective multi-agent structures, thereby addressing primarily the structural optimization dimension. In summary, while these works have advanced non-gradient MAS optimization from various angles, they typically address only a single aspect, either functional or structural, or use supervised signals indirectly within evolutionary search algorithms like MCTS. OMAC, by contrast, provides a unified framework that jointly optimizes both functionality and collaboration structure, employing direct contrastive reasoning with supervised signals rather than relying on evolutionary or search-based methods.

## 6 CONCLUSION

In this study, we introduce OMAC, a unified framework for optimizing LLM-based multi-agent systems in multi-step collaboration. We identify and formalize five key optimization dimensions addressing both agent functionality and collaboration structure. Based on them, we develop a general algorithm for individually optimizing each dimension. The algorithm leverages two LLM-powered actors to explore diverse semantic possibilities for instructing agents or controllers, and to exploit supervised contrastive pairs for refining MAS through contrastive reasoning. Furthermore, we propose an iterative algorithm for jointly optimizing multiple dimensions. Extensive experiments demonstrate the effectiveness and superiority of our optimization framework and the proposed algorithms.

## REPRODUCIBILITY STATEMENT

We provide anonymized code at https://anonymous.4open.science/r/OMAC-Sub-3FF8 , together with the datasets used in our experiments. Comprehensive details of the experimental setup, including hyperparameters, training details, and evaluation methods, are presented in the Experiments section and Appendix. With these resources, we are confident that readers will be able to reproduce the results presented in the paper.

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

## A    OPTIMIZATION DIMENSION DETAILS

We first articulate the intuition and motivation underlying the proposal of these five dimensions. As outlined in Section 3.1, they are deliberately designed to capture the fundamental aspects of MAS optimization. While task-dependent considerations such as computational efficiency may arise, our proposed dimensions highlight the core, generalizable elements of multi-agent collaboration, which constitute the central conceptual contribution of this work.

Specifically, we conceptualize the collaboration process in MAS as information flow over a graph, where agents correspond to nodes and communication channels to edges. Within this abstraction, the two functional dimensions focus on optimizing node-level capabilities, either by enhancing existing agents or constructing new ones. On the other hand, the three structural dimensions pertain to graph structure design: they govern the mechanisms of global and dynamic local agent selection, as well as the configuration of inter-agent communication pathways.

Taken together, these five dimensions form a comprehensive framework that addresses both the functional and structural foundations of MAS. By doing so, they provide principled coverage of the essential components required for optimizing the multi-agent collaboration graph.

We next present more details of the rationale and implementation details of the five dimensions.

**Fun-1: Optimizing existing agents.** This dimension focuses on enhancing the functionality of an existing agent within a MAS. Specifically, we optimize the agent's instruction prompt and/or associated few-shot examples (in few-shot learning settings) following the optimization procedure detailed in Section 3.2.

When optimizing this dimension, the input to the Semantic Initializer includes the task description and the functional role of the agent. We also provide the original instruction prompt and/or associated few-shot examples of the agent being optimized as a one-shot example for the expected output. For the Contrastive Comparator, the input context consists of the task description, the functional illustration of the agent, and the sampled positive-negative pair. Both the Semantic Initializer and the Contrastive Comparator output newly generated prompts and/or few-shot examples to power the targeted agent.

**Fun-2: Optimizing construction of new agents.** This dimension aims to optimize the design and construction of a new agent, which is subsequently integrated into the existing MAS to enhance task performance. Specifically, we generate and optimize the instruction prompt and/or associated few-shot examples used to guide a new LLM-based agent which will be incorporated into existing MAS for collaboration.

During optimization, the Semantic Initializer receives the task description and the functional roles of all existing agents in the MAS as input context. We then provide a handcrafted instruction prompt as a one-shot example of the expected output from the Semantic Initializer. For the Contrastive Comparator, the input context includes the task description along with the sampled positive-negative pair. Both the Semantic Initializer and Contrastive Comparator generate the instruction prompts and/or associated few-shot examples powering the new agent.

**Str-1: Optimizing candidate agent selection.** This dimension focuses on optimizing an LLM-powered controller to select appropriate candidate agents from all available agents before collaboration begins. Specifically, we optimize the instruction prompt that guides the controller's selection process based on the provided task context and the functional roles of all available agents.

During optimization of it, the Semantic Initializer receives as input the task description and the expected functionality of the controller. Also, a handcrafted instruction prompt is given as a one-shot example. For the Contrastive Comparator, the input context consists of the task description, the controller's expected functionality, and the sampled positive-negative pair. Both the Semantic Initializer and Contrastive Comparator produce the instruction prompts for guiding the controller in candidate agent selection.

The rationale for optimizing this controller is to selectively incorporate only those agents most beneficial for resolving the given task, thus reducing harmful disturbance and enhancing the overall effectiveness and efficiency of the MAS.

**Str-2: Optimizing dynamic agent participation.** This dimension targets to optimize an LLM-powered controller that dynamically selects agents from the candidate pool for participation in the current collaboration step. Specifically, the controller's instruction prompt is optimized to enable it to choose the most suitable agents based on the task context, the functional roles of candidate agents, and the agents' outputs generated in previous steps.

During optimization for this dimension, the input to the Semantic Initializer includes the task description, the expected functionality of the controller, and a handcrafted instruction prompt as a one-shot example. Similarly, the input context for the Contrastive Comparator comprises the task description, the controller's intended functionality, and the sampled positive-negative pair. Both the Semantic Initializer and Contrastive Comparator generate prompts instructing the controller to dynamically select appropriate agents.

The rationale for optimizing this controller is to enable the selection of only those agents most relevant and beneficial for the current collaboration step, based on analysis of the task context, agent functionality, and agents' outputs from previous steps. For instance, an agent responsible for initial problem decomposition would ideally be selected by this controller only during the early stages of the collaboration.

**Str-3: Optimizing agent communication flows.** This dimension is designed to optimize an LLM-powered controller responsible for determining whether the output from one agent should be incorporated as input context for another agent during collaboration. Specifically, we optimize the controller's instruction prompt to guide communication-structure decisions based on the given task context and the functional roles of candidate agents.

During the optimization procedure for this dimension, the input provided to the Semantic Initializer includes the task description, the expected functionality of the controller, and a handcrafted instruction prompt as a one-shot example. For the Contrastive Comparator, the input context comprises the task description, the controller's intended functionality, and the sampled positive-negative pair. Both the Semantic Initializer and the Contrastive Comparator generate prompts instructing the controller to manage communication flows between agents.

The rationale behind optimizing this controller is to route information selectively, ensuring that an agent receives input only from other agents whose outputs are directly relevant and beneficial to the recipient agent's task resolution. For example, in the code-generation task, the output from an agent serving as a "Tester," which generates and executes unit tests to evaluate previously generated code, should ideally be routed by the controller only to agents responsible for further refining the code further, rather than to another "Tester" agent.

# B    EXPERIMENT DETAILS AND ADDITIONAL RESULTS

## B.1    EXPERIMENTAL SETUP

As outlined in Section 4, we inherit most of the experimental settings from the SOTA method DyLAN Liu et al. (2024), as these configurations are standard and widely validated in existing literature for direct and fair comparison. Specifically, for all datasets, we partition the data into training and testing sets using a 1:1 ratio. We directly inherited most hyperparameters from DyLAN to maintain fair comparisons and, given limited computational resources, did not perform additional tuning on a validation set. The maximum token length is set to 2048 for code generation and arithmetic reasoning tasks, and 1024 for general reasoning tasks. The ranking and selection procedures for controllers optimized under Str-1 and Str-2 follow a listwise approach. To avoid positional bias, agent messages from the preceding collaboration step are randomly shuffled before being passed to agents in the subsequent step. Additionally, we employ an early-stopping mechanism to reduce unnecessary computational costs when agent outputs remain consistent across consecutive steps. Further details are available in the original DyLAN paper Liu et al. (2024). All experiments are repeated three times, and the mean and standard deviation are reported.

**Experiments on general reasoning tasks.** We randomly sample 68 multiple-choice questions from the MMLU dataset Hendrycks et al. (2021a) and evenly split them into training and testing sets. The default MAS comprises seven agents with the following functional roles: "Economist", "Doctor", "Lawyer", "Mathematician", "Psychologist", "Programmer", "Historian". The default instruction

prompts for these agents are adopted directly from DyLAN's implementation. Answers for the agents are extracted from the agent outputs by matching the final occurrence of "(X" or "(X)", where "X" denotes one of the choices A, B, C, or D. The final answer is determined by selecting the option that receives the highest number of votes from agents in the last step of collaboration. The maximum number of collaboration steps is set to four, with dynamic agent selection occurring at the third step. Performance is always measured by the average classification accuracy across all questions in the four categories.

**Experiments on code generation tasks.** We sample 80 function-level code completion tasks from the HumanEval benchmark Chen et al. (2021) and evenly divide them into training and testing sets. Following DyLAN, the default MAS comprises four code-writing agents and four code-reviewing agents. Among code reviewers, three are optimizable while the fourth remains fixed as required by the method workflow. The code writers include "Python Assistant", "Algorithm Developer", "Computer Scientist", and "Programmer". The optimizable code reviewers are "Syntax Checker", "Unit Tester", and "Reflector". Solutions provided by code writers undergo a review process with a maximum of six rounds. Specifically, at time steps $t = 1, 3, 4, 6$, code writers generate solutions, while at $t = 2, 5$, code reviewers provide feedback. Dynamic agent selection occurs at the fourth step. The final code is randomly selected from the top five code completions across all agents that pass most tests from code reviewers. Performance evaluation is based on the average pass rate (Pass@1) of generated code across all code completion tasks.

**Experiments on arithmetic reasoning tasks.** We randomly sample 140 mathematical problems from the MATH dataset Hendrycks et al. (2021b), evenly covering seven subareas: algebra, counting and probability, geometry, intermediate algebra, number theory, pre-algebra, and pre-calculus. Also, these samples are split evenly into training and testing sets. In DyLAN, the authors found that collaborating agents in different domains (e.g., algebra and geometry experts) do not make significant improvement, therefore they adopt four agents with identical prompts and few-shot examples. We follow this setting for a fair comparison. The maximum number of collaboration steps is set to four, with dynamic agent selection taking place at the third step. We also follow DyLAN in using the answer extraction method described by Hendrycks et al. (2021b). The average accuracy across all questions serves as the performance evaluation.

### B.2 ADDITIONAL EXPERIMENTAL RESULTS

#### B.2.1 SENSITIVITY OF HYPER-PARAMETERS FOR SINGLE-DIMENSION OPTIMIZATION

**Size of initial collection.** We first evaluate the hyper-parameter sensitivity by varying the size of initial collection (denoted as $z$) generated by the Semantic Initializer, while keeping all other settings as default. Table 5 summarizes the results on the MATH dataset for arithmetic reasoning tasks. The results demonstrate that increasing the size of the initial collection generally leads to improved performance with OMAC. This is intuitively reasonable because allowing the Semantic Initializer to explore more diverse agent/controller designs increases the likelihood of obtaining higher-performing solutions. While a larger collection entails greater computational costs since each initialized agent/controller must be evaluated via the full multi-agent collaboration on the training data, we observe substantial improvements over the SOTA DyLAN even with a modest collection size of three.

Table 5: Accuracy (%) of OMAC on each dimension with different sizes of initial collection on arithmetic reasoning tasks.

| Collection Size | Str-1 | Str-2 | Str-3 | Fun-1.1 | Fun-1.2 | Fun-2 |
|:---:|:---:|:---:|:---:|:---:|:---:|:---:|
| $z = 1$ | $32.71_{\pm 1.63}$ | $32.75_{\pm 1.98}$ | $32.80_{\pm 2.32}$ | $33.19_{\pm 2.34}$ | $32.98_{\pm 2.82}$ | $32.80_{\pm 2.14}$ |
| $z = 2$ | $33.14_{\pm 1.50}$ | $32.97_{\pm 1.41}$ | $33.25_{\pm 2.65}$ | $34.46_{\pm 2.40}$ | $33.90_{\pm 2.15}$ | $33.33_{\pm 2.49}$ |
| $z = 3$ | $33.34_{\pm 1.45}$ | $33.41_{\pm 1.37}$ | $33.70_{\pm 1.62}$ | $35.17_{\pm 1.96}$ | $34.91_{\pm 2.03}$ | $33.95_{\pm 1.30}$ |
| $z = 4$ | $33.69_{\pm 1.53}$ | $33.73_{\pm 1.42}$ | $33.96_{\pm 1.27}$ | $35.63_{\pm 1.52}$ | $35.21_{\pm 1.72}$ | $34.17_{\pm 1.53}$ |
| $z = 5$ | $33.92_{\pm 1.23}$ | $33.96_{\pm 0.96}$ | $34.25_{\pm 1.27}$ | $35.89_{\pm 2.14}$ | $35.67_{\pm 1.42}$ | $34.51_{\pm 1.31}$ |

**Maximum number of contrasting iterations.** We then conduct experiments to assess the impact of the maximum number of contrastive reasoning iterations (denoted as $w$) used by the Contrastive Comparator. Table 6 presents the results on the MATH dataset for arithmetic reasoning tasks. The

results indicate a similar trend: increasing the number of iterations generally results in enhanced performance by consistently refining the agents/controllers through contrastive reasoning. Furthermore, setting the iteration count to three is sufficient to achieve significant performance improvements compared to DyLAN.

Table 6: Accuracy (%) of OMAC on each dimension with different maximum number of contrastive reasoning iterations on arithmetic reasoning tasks.

| Number of Iterations | Str-1 | Str-2 | Str-3 | Fun-1.1 | Fun-1.2 | Fun-2 |
|---|---|---|---|---|---|---|
| $w = 1$ | $32.84_{\pm1.23}$ | $32.72_{\pm2.73}$ | $32.83_{\pm2.41}$ | $33.45_{\pm2.23}$ | $33.03_{\pm2.31}$ | $32.82_{\pm2.16}$ |
| $w = 2$ | $33.02_{\pm2.44}$ | $32.94_{\pm2.14}$ | $33.07_{\pm2.61}$ | $34.12_{\pm2.24}$ | $34.12_{\pm2.60}$ | $33.43_{\pm2.57}$ |
| $w = 3$ | $33.34_{\pm1.45}$ | $33.41_{\pm1.37}$ | $33.70_{\pm1.62}$ | $35.17_{\pm1.96}$ | $34.91_{\pm2.03}$ | $33.95_{\pm1.30}$ |
| $w = 4$ | $33.85_{\pm1.23}$ | $34.15_{\pm1.14}$ | $34.41_{\pm1.52}$ | $35.74_{\pm1.12}$ | $35.26_{\pm0.67}$ | $34.34_{\pm0.93}$ |
| $w = 5$ | $34.05_{\pm0.84}$ | $34.77_{\pm0.73}$ | $34.82_{\pm1.14}$ | $36.04_{\pm1.62}$ | $35.62_{\pm1.26}$ | $34.79_{\pm1.52}$ |

**Sampling thresholds.** Lastly, We evaluate the impact of varying sampling thresholds $l$ and $h$, as described in Section 3.2. Table 6 presents the results for dimensions Str-1 and Fun-1.1 on the MATH dataset for arithmetic reasoning tasks. From these results, we observe that OMAC is robust to variations in the sampling thresholds, with performance fluctuations limited to within 1%. Additionally, we note a slight performance decrease when the gap between $l$ and $h$ widens. This may be attributed to imbalanced sampling of positive and negative examples, potentially leading to less accurate and fair reasoning by the Contrastive Comparator.

Table 7: Accuracy (%) of OMAC on Str-1 (left) and Fun-1.1 (right) with different combinations of sampling thresholds on arithmetic reasoning tasks.

| | $l = 0.3$ | $l = 0.4$ | $l = 0.5$ | | $l = 0.3$ | $l = 0.4$ | $l = 0.5$ |
|---|---|---|---|---|---|---|---|
| $h = 0.3$ | 32.94 | 33.02 | 32.86 | $h = 0.3$ | 34.92 | 34.95 | 34.90 |
| $h = 0.4$ | 32.90 | 33.18 | 33.10 | $h = 0.4$ | 35.02 | 35.01 | **35.19** |
| $h = 0.5$ | 32.79 | **33.41** | 33.34 | $h = 0.5$ | 34.87 | 35.10 | 35.17 |

### B.2.2 Additional Results of Ablation Study

Tables 8 and Table 9 summarize the results of the ablation study described in Section 4.3 on code generation and general reasoning tasks. The findings are consistent: the ablation model, OMAC-C, which removes the Contrastive Comparator from the optimization pipeline, performs significantly worse than the full OMAC framework. These results highlight the substantial advantage provided by the Contrastive Comparator's contrastive reasoning on supervised positive-negative pairs.

Table 8: Pass@1 (%) of OMAC-C and OMAC with each optimization dimension on the code generation tasks.

| Method | Str-1 | Str-2 | Str-3 |
|---|---|---|---|
| OMAC-C | $86.23_{\pm1.67}$ | $85.66_{\pm2.34}$ | $86.31_{\pm2.80}$ |
| OMAC | $86.76_{\pm1.22}$ | $86.92_{\pm2.27}$ | $87.55_{\pm2.46}$ |

| Method | Fun-1.1 | Fun-1.2 | Fun-1.3 | Fun-1.4 | Fun-1.5 | Fun-1.6 | Fun-1.7 | Fun-2 |
|---|---|---|---|---|---|---|---|---|
| OMAC-C | $86.74_{\pm2.96}$ | $85.92_{\pm2.42}$ | $86.46_{\pm2.21}$ | $87.86_{\pm1.88}$ | $87.11_{\pm3.56}$ | $87.26_{\pm2.04}$ | $86.97_{\pm1.95}$ | $86.01_{\pm2.86}$ |
| OMAC | $88.39_{\pm2.54}$ | $86.31_{\pm2.21}$ | $88.87_{\pm1.36}$ | $89.25_{\pm1.30}$ | $88.74_{\pm2.67}$ | $88.39_{\pm1.22}$ | $88.34_{\pm1.42}$ | $86.77_{\pm2.43}$ |

### B.2.3 Additional Results of Multi-Dimension Optimization

**Results on other tasks.** We present experimental results of multi-dimension optimization using our OMAC on code generation tasks and general reasoning tasks in Figure 5 and Figure 6 respectively. The trends are similar to the results shown in Section 4.2, which demonstrate that iterative optimizing multiple dimensions can bring in **significant performance improvements compared to optimizing**

Table 9: Accuracy (%) of OMAC-C and OMAC with each optimization dimension on the general reasoning tasks.

| Method | Str-1 | Str-2 | Str-3 |
|---|---|---|---|
| OMAC-C | $71.13_{\pm1.94}$ | $69.86_{\pm1.77}$ | $70.71_{\pm1.75}$ |
| OMAC | $73.14_{\pm2.24}$ | $72.06_{\pm1.96}$ | $73.18_{\pm1.47}$ |

| Method | Fun-1.1 | Fun-1.2 | Fun-1.3 | Fun-1.4 | Fun-1.5 | Fun-1.6 | Fun-1.7 | Fun-2 |
|---|---|---|---|---|---|---|---|---|
| OMAC-C | $70.88_{\pm2.55}$ | $70.47_{\pm2.04}$ | $70.46_{\pm2.74}$ | $71.70_{\pm2.15}$ | $71.33_{\pm3.10}$ | $70.19_{\pm2.35}$ | $70.23_{\pm2.05}$ | $69.74_{\pm2.93}$ |
| OMAC | $72.33_{\pm2.47}$ | $73.23_{\pm1.83}$ | $72.06_{\pm2.41}$ | $74.22_{\pm2.22}$ | $73.15_{\pm2.86}$ | $72.07_{\pm2.92}$ | $71.83_{\pm2.74}$ | $71.02_{\pm2.57}$ |

**a single dimension**. Also, only optimizing dimensions that individually demonstrate the most significant improvements is an effective strategy to bring in significant performance improvement considering the computation resource constraints.

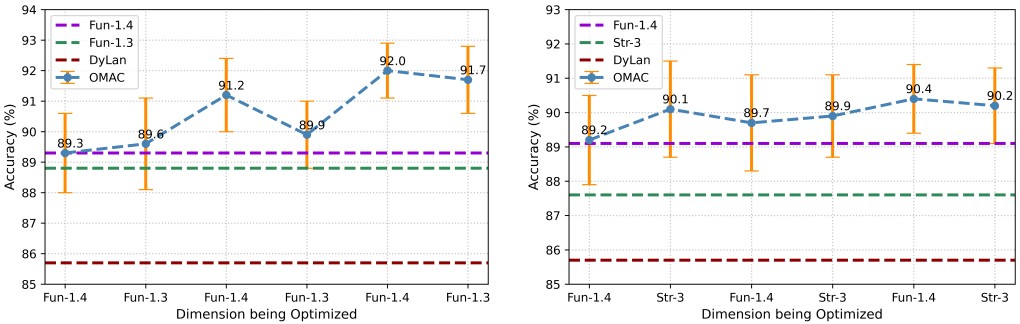

Figure 5: Performance during iterative optimization for multiple dimensions on code generation tasks. The X-axis represents the dimensions undergoing three iterations of optimization, while each point indicates the MAS's performance with the optimized dimensions on the test set. The error bar denotes the standard deviation.

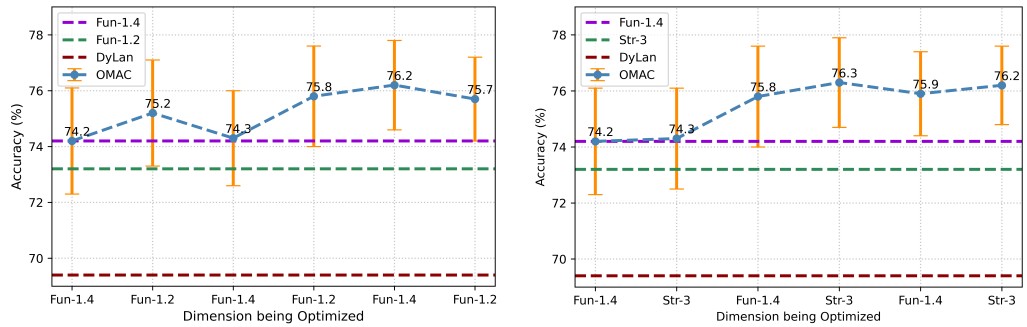

Figure 6: Performance during iterative optimization for multiple dimensions on general reasoning tasks. The X-axis represents the dimensions undergoing three iterations of optimization, while each point indicates the MAS's performance with the optimized dimensions on the test set. The error bar denotes the standard deviation.

**Validation of iterative optimization.** We further conduct experiments to validate our iterative optimization design when jointly optimizing multiple dimensions. Specifically, we propose to optimize one dimension at a time while keeping other dimensions fixed, thereby preserving the effectiveness of contrastive reasoning by limiting variability within positive-negative pairs. To verify this, we conduct experiments where two dimensions are simultaneously varied during optimization,

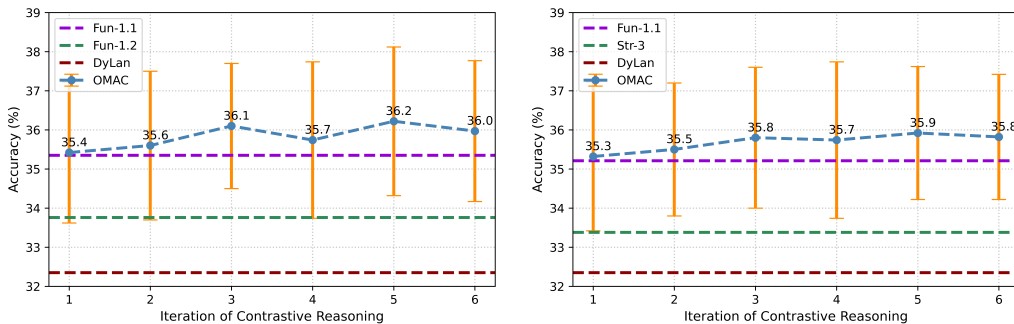

Figure 7: Performance of simultaneously optimizing multiple dimensions on arithmetic reasoning tasks. The X-axis denotes the iteration number of contrastive reasoning, and each point indicates the MAS performance achieved using the optimized dimensions generated by the Contrastive Comparator. The left figure illustrates results from jointly optimizing Fun-1.1 and Fun-1.2, while the right figure corresponds to jointly optimizing Fun-1.1 and Str-3. The error bar denotes the standard deviation.

meaning the Contrastive Comparator generates two agents/controllers based on the positive-negative pair in each iteration.

By comparing Figure 7 and Figure 4, it is evident that simultaneously optimizing multiple dimensions, which asks the Contrastive Comparator to reason over multiple variable factors at once, results in significantly reduced performance gains and larger variance for OMAC. These findings validate the rationale behind our iterative multi-dimensional optimization design.

### B.2.4 RESULTS WITH DIFFERENT BASE LLMS

In addition to using GPT-3.5-Turbo as the base model, we also conducted experiments with three other LLMs: GPT-4o-mini and two open-source models, DeepSeek-V2.5 and Qwen-2.5-72B. The corresponding results are reported separately in Table 10, Table 11, and Table 12. The findings consistently demonstrate OMAC's robustness and its sustained performance advantages over baseline methods across all optimization dimensions. These results further highlight the cross-model transferability of the optimization capabilities provided by the proposed OMAC framework.

Table 10: Performance on code generation tasks with GPT-4o-mini.

| Method | Baselines | | | | | OMAC (Structural Dimension) | | |
|---|---|---|---|---|---|---|---|---|
| | CodeT | AgentVerse | ADAS | AFlow | DyLAN | Str-1 | Str-2 | Str-3 |
| Pass@1 | $79.24_{\pm2.14}$ | $84.72_{\pm1.95}$ | $83.83_{\pm2.04}$ | $87.31_{\pm2.70}$ | $\underline{87.56_{\pm2.13}}$ | $\mathbf{88.69_{\pm1.67}}$ | $\mathbf{88.73_{\pm1.74}}$ | $\mathbf{89.08_{\pm1.97}}$ |

| Method | OMAC (Functional Dimension) | | | | | | | |
|---|---|---|---|---|---|---|---|---|
| | Fun-1.1 | Fun-1.2 | Fun-1.3 | Fun-1.4 | Fun-1.5 | Fun-1.6 | Fun-1.7 | Fun-2 |
| Pass@1 | $\mathbf{89.30_{\pm1.41}}$ | $\mathbf{88.78_{\pm1.64}}$ | $\mathbf{88.95_{\pm1.80}}$ | $\mathbf{90.42_{\pm1.75}}$ | $\mathbf{89.43_{\pm2.06}}$ | $\mathbf{88.60_{\pm1.42}}$ | $\mathbf{89.36_{\pm1.04}}$ | $\mathbf{88.45_{\pm0.97}}$ |

### B.2.5 EXPERIMENTS ON OTHER BENCHMARKS

Firstly, we'd like to highlight that the benchmarks we've chosen in the main paper (HumanEval, MMLU, MATH) are well-established in the MAS literature and widely utilized by recent prominent studies such as DyLAN Liu et al. (2024), ADAS Hu et al. (2025), LLM Debate Du et al. (2023), and Agentverse Chen et al. (2024a), for evaluating both multi-turn MAS and their optimization methods. Thus, we consider these benchmarks sufficiently representative to evaluate MAS performance, ensuring that our results are reliable and comparable with existing literature.

Nevertheless, to substantiate evaluate the effectiveness and generalizability of OMAC on more challenging and complex scenarios, we conducted additional experiments on the on the MBPP Austin

Table 11: Performance on code generation tasks with DeepSeek-V2.5.

| Method | Baselines | | | | | OMAC (Structural Dimension) | | |
|---|---|---|---|---|---|---|---|---|
| | CodeT | AgentVerse | ADAS | AFlow | DyLAN | Str-1 | Str-2 | Str-3 |
| Pass@1 | $79.88_{\pm1.52}$ | $85.11_{\pm2.62}$ | $83.98_{\pm2.69}$ | $\underline{87.95}_{\pm2.06}$ | $87.74_{\pm1.95}$ | $\mathbf{89.21}_{\pm1.76}$ | $\mathbf{88.92}_{\pm2.31}$ | $\mathbf{89.37}_{\pm1.47}$ |

| Method | OMAC (Functional Dimension) | | | | | | | |
|---|---|---|---|---|---|---|---|---|
| | Fun-1.1 | Fun-1.2 | Fun-1.3 | Fun-1.4 | Fun-1.5 | Fun-1.6 | Fun-1.7 | Fun-2 |
| Pass@1 | $\mathbf{89.90}_{\pm2.52}$ | $\mathbf{88.95}_{\pm1.92}$ | $\mathbf{89.35}_{\pm1.52}$ | $\mathbf{90.35}_{\pm1.47}$ | $\mathbf{89.74}_{\pm1.67}$ | $\mathbf{89.21}_{\pm2.04}$ | $\mathbf{89.90}_{\pm1.81}$ | $\mathbf{89.02}_{\pm1.74}$ |

Table 12: Performance on code generation tasks with Qwen-2.5-72B.

| Method | Baselines | | | | | OMAC (Structural Dimension) | | |
|---|---|---|---|---|---|---|---|---|
| | CodeT | AgentVerse | ADAS | AFlow | DyLAN | Str-1 | Str-2 | Str-3 |
| Pass@1 | $79.27_{\pm2.68}$ | $84.14_{\pm2.41}$ | $83.29_{\pm2.67}$ | $86.95_{\pm2.49}$ | $\underline{87.02}_{\pm2.72}$ | $\mathbf{88.32}_{\pm1.97}$ | $\mathbf{88.56}_{\pm2.15}$ | $\mathbf{88.91}_{\pm1.87}$ |

| Method | OMAC (Functional Dimension) | | | | | | | |
|---|---|---|---|---|---|---|---|---|
| | Fun-1.1 | Fun-1.2 | Fun-1.3 | Fun-1.4 | Fun-1.5 | Fun-1.6 | Fun-1.7 | Fun-2 |
| Pass@1 | $\mathbf{89.26}_{\pm1.87}$ | $\mathbf{88.83}_{\pm2.41}$ | $\mathbf{88.74}_{\pm1.77}$ | $\mathbf{90.02}_{\pm1.65}$ | $\mathbf{89.13}_{\pm1.98}$ | $\mathbf{88.64}_{\pm1.83}$ | $\mathbf{88.95}_{\pm2.01}$ | $\mathbf{88.16}_{\pm1.33}$ |

et al. (2021) benchmark and GAIA Mialon et al. (2023) benchmark. MBPP benchmark consists of crowd-sourced Python programming tasks designed to emulate more complex and realistic problem-solving scenarios encountered by entry-level programmers. Similarly, we adopt DyLAN's setup on the HumanEval benchmark as the default setting for OMAC, since both benchmarks focus on code generation tasks. As shown in Table 13, the results confirm OMAC's superior performance compared to most recent MAS optimization frameworks, including DyLAN Liu et al. (2024) and AFLOW Zhang et al. (2025c), on more challenging tasks.

The GAIA benchmark consists of challenging real-world questions that require a broad set of fundamental capabilities, including reasoning, multi-modal understanding, web browsing, and general tool-use proficiency. Following prior work on similar tasks Yao et al. (2022); Huang et al. (2023), we adopt a standard set of agents for tool-using tasks as our default configuration. Specifically, the default agent set includes Self-Consistency, Chain-of-Thought, LLM-Debate, Testing, Ensemble, Self-Refine, and ReAct. As shown in Table 14, OMAC achieves superior performance compared to existing MAS and MAS optimization approaches on this challenging tool-calling benchmark.

Table 13: Performance on MBPP benchmark.

| Method | Baselines | | | | OMAC (Structural Dimension) | | |
|---|---|---|---|---|---|---|---|
| | CodeT | CAMEL | Aflow | DyLAN | Str-1 | Str-2 | Str-3 |
| Pass@1(%) | $65.72_{\pm2.04}$ | $70.43_{\pm2.41}$ | $81.45_{\pm1.98}$ | $79.32_{\pm2.03}$ | $\mathbf{82.81}_{\pm1.87}$ | $\mathbf{82.50}_{\pm1.69}$ | $\mathbf{83.23}_{\pm2.35}$ |

| Method | OMAC (Functional Dimension) | | | | | | | |
|---|---|---|---|---|---|---|---|---|
| | Fun-1.1 | Fun-1.2 | Fun-1.3 | Fun-1.4 | Fun-1.5 | Fun-1.6 | Fun-1.7 | Fun-2 |
| Pass@1(%) | $\mathbf{82.49}_{\pm2.81}$ | $\mathbf{83.30}_{\pm2.45}$ | $\mathbf{82.97}_{\pm1.84}$ | $\mathbf{83.02}_{\pm1.48}$ | $\mathbf{83.43}_{\pm1.60}$ | $\mathbf{83.11}_{\pm1.73}$ | $\mathbf{83.06}_{\pm1.36}$ | $\mathbf{82.35}_{\pm1.55}$ |

### B.2.6 COMPUTATION COST

First, OMAC improves online inference efficiency, which is a key factor for the practical deployment of agent systems, through its effective structural control. Specifically, OMAC optimizes to dynamically select only the most important and valuable agents to participate in the collaboration (Str-1 and Str-2) and intelligently determines the input context by incorporating only relevant information

Table 14: Performance on GAIA benchmark (Level 2).

| Method | Baselines | | | | OMAC (Structural Dimension) | | |
|---|---|---|---|---|---|---|---|
| | AutoGPT | ADAS | Aflow | DyLAN | Str-1 | Str-2 | Str-3 |
| Pass@1(%) | $1.04_{\pm 0.15}$ | $3.87_{\pm 1.25}$ | $8.23_{\pm 2.63}$ | $9.82_{\pm 2.41}$ | $\mathbf{11.93}_{\pm 2.06}$ | $\mathbf{10.16}_{\pm 1.97}$ | $\mathbf{12.88}_{\pm 1.49}$ |

| Method | OMAC (Functional Dimension) | | | | | | | |
|---|---|---|---|---|---|---|---|---|
| | Fun-1.1 | Fun-1.2 | Fun-1.3 | Fun-1.4 | Fun-1.5 | Fun-1.6 | Fun-1.7 | Fun-2 |
| Pass@1(%) | $\mathbf{12.14}_{\pm 2.81}$ | $\mathbf{10.92}_{\pm 2.45}$ | $\mathbf{11.73}_{\pm 1.84}$ | $\mathbf{13.04}_{\pm 1.48}$ | $\mathbf{11.97}_{\pm 1.60}$ | $\mathbf{13.74}_{\pm 1.73}$ | $\mathbf{10.70}_{\pm 1.36}$ | $\mathbf{14.24}_{\pm 2.55}$ |

(Str-3). These optimizations substantially reduce token consumption and API calls. We report the average number of API calls and token cost with GPT-3.5-Turbo required to solve a single problem on the code generation task (HumanEval benchmark) and the arithmetic reasoning task (MATH dataset), shown in Table 15 and Table 16. The results demonstrate that OMAC significantly reduces inference time computational cost by dynamically optimizing the collaboration structure, outperforming other multi-agent frameworks.

Table 15: Inference cost on code generation tasks.

| Method | Baselines | | | OMAC | | | |
|---|---|---|---|---|---|---|---|
| | CodeT | AgentVerse | DyLAN | Str-1 | Str-2 | Str-3 | Str-2+Str-3 |
| #API Calls | 20.72 | 23.15 | 17.94 | 16.34 | 15.76 | 17.03 | 14.80 |
| Cost ($) | 0.1430 | 0.1958 | 0.1288 | 0.1032 | 0.0973 | 0.0920 | 0.0883 |

Table 16: Inference cost on arithmetic reasoning tasks.

| Method | Baselines | | | OMAC | | | |
|---|---|---|---|---|---|---|---|
| | LLM-Blender | LLM-Debate | DyLAN | Str-1 | Str-2 | Str-3 | Str-2+Str-3 |
| #API Calls | 7.12 | 8.26 | 7.30 | 6.85 | 6.23 | 7.12 | 5.74 |
| Cost ($) | 0.0458 | 0.0530 | 0.0446 | 0.0369 | 0.0352 | 0.0337 | 0.0316 |

For the cost on the training stage, as illustrated in Section 3.2, OMAC requires evaluating agents or controllers by executing the full collaborative process across the training dataset. While this approach ensures robust and representative performance evaluations, it can increase computational costs especially when the size of training data is large and the given tasks are complex. For example, OMAC requires around 1,400 API calls for training to optimize each dimension on the HumanEval benchmark.

However, we'd like to highlight that several recent studies optimizing agent systems in a supervised and iterative manner for task-level performance, such as AVATAR Wu et al. (2024) and PPDPP Deng et al. (2023), also impose similar computational demands, often involving thousands of API calls during system optimization. For instance, AVATAR, which is designed to optimize only a single agent, requires over 3,000 API calls for just one training round on the FLICKR30K ENTITIES dataset Plummer et al. (2015). Therefore, we regard these computational constraints as a common challenge across current agent system optimization methods, rather than a limitation unique to OMAC.

Nevertheless, balancing OMAC's effectiveness with computational efficiency remains a valuable and promising direction for the practical implementation of OMAC, particularly for academic researchers with limited computational resources. To investigate this, we propose a simple yet effective strategy: evaluating candidate agents or controllers on randomly sampled subsets of the training data. Although this may introduce variance, it provides a good starting point for exploring the trade-off between efficiency and robustness.

Specifically, we conducted experiments on the HumanEval benchmark, where only 50% of the training data was randomly sampled for each evaluation. This approach reduced the number of API

calls per optimization by approximately 50%, lowering the total cost of a complete training run with GPT-3.5-turbo to around 5 dollars. Despite the reduced evaluation set, OMAC continued to outperform the strongest baseline, DyLAN, with only a minor decrease in performance compared to evaluations conducted on the full training set. These findings suggest that subset-based evaluation offers a practical and effective way to balance computational efficiency and evaluation robustness for optimization in OMAC.

Table 17: Training cost on code generation tasks with sampled training data.

| Method | Baseline | OMAC | | | | | | | | | | |
|---|---|---|---|---|---|---|---|---|---|---|---|---|
| | DyLAN | Fun-1.1 | Fun-1.2 | Fun-1.3 | Fun-1.4 | Fun-1.5 | Fun-1.6 | Fun-1.7 | Fun-2 | Str-1 | Str-2 | Str-3 |
| Accuracy | 85.74 | 88.27 | 86.22 | 88.62 | 89.13 | 88.51 | 88.20 | 88.23 | 86.56 | 86.43 | 86.79 | 87.47 |
| Cost ($) | 4.62 | 4.95 | 5.21 | 5.02 | 5.40 | 5.67 | 5.28 | 5.15 | 4.73 | 4.62 | 5.41 | 5.24 |

Last, we include a comprehensive comparison on the training cost, inference cost, and performance of OMAC with the main baselines including AgentVerse, DyLAN, and AFlow. The experiments are also conducted on the HumanEval benchmark, and we report the average cost and performance across all optimization dimensions of OMAC for comparison.

Table 18: Efficiency comparison between OMAC's average performance and other MAS baselines on code generation tasks.

| Method | AgentVerse | DyLAN | AFlow | OMAC (avg.) |
|---|---|---|---|---|
| Training Cost ($) | - | **4.62** | 8.24 | 5.15 |
| Inference Cost ($) | 0.1928 | 0.1288 | 0.1134 | **0.0974** |
| Accuracy (%) | 78.26 | 85.74 | 85.63 | **87.68** |

From Table 18, it's clear that OMAC achieves substantial improvements in both inference efficiency and performance than all compared baselines, with only a minor increase in training cost compared to DyLAN. These findings further confirm OMAC's strong cost–performance balance and practical applicability for real-world multi-agent systems.

### B.2.7 TRAINING CURVE

We further evaluate OMAC's performance after each contrastive reasoning step when iteratively optimizing two dimensions (Fun1.2 then Fun1.1) over four iterations on arithmetic reasoning tasks. In particular, we report the performance curves on both the training set and the test set in Figure 8.

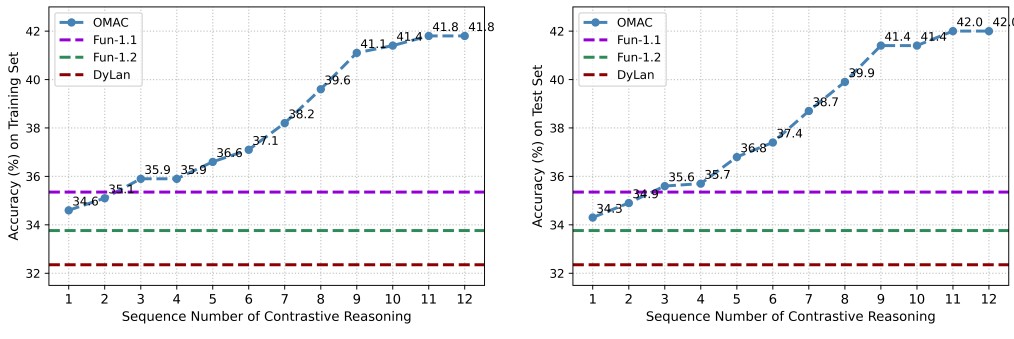

Figure 8: Performance after each contrastive reasoning on arithmetic reasoning tasks. The X-axis represents the sequence number of contrastive reasoning during four iterations of optimization, while each point indicates the MAS's performance with the optimized dimensions on the training set (left) and the test set (right).

Although OMAC's optimization is guided by performance on a fixed training set, the results reveal a strong alignment between training and test performance, indicating that no overfitting occurs. We

attribute this robustness to two key design choices. First, the optimization dimensions are constructed to target generalizable, task-level instructions rather than narrowly tailored prompts designed for specific cases. Second, each optimization iteration evaluates performance averaged across the entire training set, which inherently discourages over-specialization to particular subsets of data.

## C  ADDITIONAL DISCUSSIONS

### C.1  DISCUSSION ON OPTIMIZATION EFFECTIVENESS

In general, OMAC's effectiveness arises from its robust iterative optimization approach explicitly guided by supervised downstream task performance. More concretely, the Semantic Initializer is responsible for broadly exploring the semantic space by generating a diverse set of candidate agents/controllers. Meanwhile, the Contrastive Comparator focuses specifically on exploiting supervised feedback signals to refine the candidates by conducting targeted contrastive analyses iteratively.

During the iterative optimization, it is indeed possible for the Contrastive Comparator to occasionally generate candidate agents or controllers that underperform relative to the established positive samples. However, to mitigate this, OMAC incorporates a robust mechanism grounded in rigorous supervised performance evaluation. Each newly generated candidate undergoes thorough downstream task assessment leveraging the whole training set, and any candidate failing to surpass the performance of the existing positive sample is subsequently treated as a negative example. This negative example is then utilized in subsequent iterations for enhanced contrastive reasoning, systematically improving solution quality over successive rounds.

As illustrated in Section 3.2, both Semantic Initializer and Contrastive Comparator significantly depend on the reasoning capability of the underlying black-box LLM. However, we emphasize that OMAC addresses this limitation through an iterative optimization process explicitly guided by supervised performance feedback. This iterative approach systematically integrates insights from previous successes and failures, allowing the LLM to progressively improve its reasoning beyond single-round capabilities.

Our empirical results demonstrate substantial and consistent performance improvements from this iterative optimization, underscoring its effectiveness beyond the baseline capabilities of the static, single-pass reasoning provided by the LLM. Also, the ablation experiments demonstrate substantial and reliable performance gains achieved through this comparator-driven iterative refinement approach. These findings highlight the practical robustness and efficacy of OMAC's iterative optimization strategy, showcasing significant improvements over baseline methods and the Semantic Initializer alone. Overall, this evidence underscores the critical importance of iterative refinement in maximizing the reasoning potential and solution quality of black-box LLM-based systems.

### C.2  DISCUSSION ON OPTIMIZATION PERFORMANCE

We present the performance improvements achieved by OMAC for single-dimension optimization in Tables 1-3, and for multi-dimension optimization in Figures 4-6. Although the average relative single-dimension optimization gains achieved by OMAC (4.6%, 2.8%, and 5.3% across the three benchmarks per dimension) may appear modest at first glance, these improvements are, in fact, significant and comparable to those reported by recent MAS optimization studies, such as DyLAN Liu et al. (2024), conducted under similar experimental conditions.

More importantly, we emphasize that OMAC's capability to simultaneously optimize multiple dimensions results in **substantially larger performance gains compared to single-dimension optimization**. For example, as demonstrated in Figures 4, the improvement in arithmetic reasoning tasks increases notably from 2.9% to 9.6% when jointly optimizing Fun-1.1 and Fun-1.2. These findings underscore the significant and practical effectiveness of OMAC's systematic multi-dimensional optimization approach in enhancing MAS performance.

### C.3  DISCUSSION ON CENTRALIZED MAS

The current OMAC framework design primarily targets flat MAS architectures, where agents are typically organized into sequential workflows to collaborate effectively. An alternative MAS architecture, known as centralized MAS Li et al. (2025); Chen et al. (2024b), features a master agent coordinating other agents for collaboration and problem-solving.

We contend that the general definitions of OMAC's optimization dimensions, including both functional and structural ones, can also effectively apply to centralized MAS frameworks. Specifically, optimization dimensions such as enhancing existing agents, developing new agents, globally and

locally selecting collaborating agents, and managing agent communication remain broadly applicable. Also, the implementation details for functional dimensions remain consistent across the two frameworks. However, structural dimensions, especially communication control, would require practical adaptations for centralized MAS, as communication typically occurs between the master agent and subordinate agents rather than between arbitrary pairs of agents. Investigating the effectiveness and detailed design adaptations of OMAC within centralized MAS settings represents a promising direction for future research.

### C.4 DISCUSSION ON BIAS ISSUE

As discussed in Section C.1, employing LLMs for contrastive reasoning may introduce biases. However, to mitigate this risk, OMAC integrates a robust iterative optimization process supported by rigorous supervised performance evaluations. Each candidate produced by the comparator is thoroughly validated against the entire training dataset, with weaker candidates systematically labeled as negative examples for subsequent iterations. This iterative refinement progressively sharpens the optimization direction.

In other words, even if a candidate from the contrastive reasoning process is biased, its adverse effect is promptly detected by subsequent evaluation and prevented from influencing later optimization stages. Instead, such biased outcomes are repurposed as negative examples in contrastive reasoning, thereby reinforcing the distinction between positive and negative cases and ultimately enhancing the overall optimization design.

However, to further enhance reasoning effectiveness, an alternative strategy could be incorporating contrastive reasoning over two batches of positive and negative samples instead of a single pair, which may enhance the generalizability of contrastive reasoning and reduce biases. But this approach also significantly increases prompt length and computational costs. Exploring the balance between reasoning effectiveness and computational efficiency represents another interesting and valuable direction for future research.

### C.5 DISCUSSION OF FUTURE WORK

While OMAC is designed as a general framework for optimizing MAS in complex tasks where empirical results validated its effectiveness, we identify several directions that warrant further exploration.

First, OMAC currently employs two LLM-based actors, the Semantic Initializer and the Contrastive Comparator, for MAS optimization. Although the existing literature on gradient-based methods targeting MAS optimization in multi-step collaboration settings remains very limited (as discussed in Section 5), exploring this direction is highly promising. Gradient-based optimization may enable more controllable, integrated, and efficient optimization processes. In particular, extending current RL-based methods for agentic optimization, such as Spiral Liu et al. (2025) or MHGPO Chen et al. (2025), into the problem setting and methodological framework of OMAC represents a valuable research avenue.

Second, tool usage has shown substantial promise and effectiveness in LLM-powered agent systems. Although our experiments on the HumanEval benchmark have already demonstrated OMAC's potential to optimize agents' tool-use abilities, extending OMAC to optimize agent collaboration and utilization across more advanced and diverse tools for solving complex real-world problems would be a significant next step. Emerging benchmarks such as GAIA Mialon et al. (2023) and SWE-bench Jimenez et al. (2023) provide suitable environments for evaluating complex tool interactions. Future studies based on these benchmarks, along with comprehensive comparisons against existing MAS approaches, could yield valuable insights.

Third, while OMAC is designed for MAS optimization across the five general dimensions we identified, our current experiments primarily focus on prompt and in-context example optimization. Other components, such as memory management or retrieval mechanisms, may also require optimization under specific task contexts. Nonetheless, these elements still conceptually fall under agent functionality or MAS structure optimization and can therefore be naturally integrated into OMAC. By leveraging LLMs' contrastive reasoning with positive/negative sample pairs, these additional

optimization tasks can be addressed following our single- and multi-dimension algorithms. Exploring OMAC's effectiveness in optimizing such components would be a valuable direction for future work.

Fourth, OMAC currently adopts an iterative optimization pattern for multi-dimension optimization. As discussed in Section 3.3 and empirically verified in Appendix B.2.3, this approach is well motivated by the necessity of maintaining effective contrastive reasoning. Nevertheless, investigating joint multi-dimension optimization remains a promising direction due to its potential for improved integration and efficiency. Methods such as gradient blending, Pareto-efficient optimization, evolutionary algorithms, data mixing, Bayesian optimization, or other multi-objective optimization techniques could be explored in combination with OMAC to enhance joint optimization.

Finally, our five proposed optimization dimensions for MAS are designed to be general and fundamental, derived from conceptualizing the MAS collaboration process as information flow over a graph, where agents correspond to nodes and communication channels to edges (see Appendix A). This paper primarily focuses on defining these dimensions and proposing the corresponding optimization methodology, rather than conducting a rigorous theoretical analysis. A deeper theoretical study of the optimization landscape, including formal guarantees of optimal configurations, would provide valuable insights and strengthen the conceptual foundation of MAS optimization research.

# D PROMPTS AND EXAMPLES

## D.1 PROMPT TEMPLATES

As described in Section 4, we adopt the agent designs and collaboration structures from the SOTA method DyLAN Liu et al. (2024) as the default configuration for OMAC on all datasets. Specifically, the default instruction prompts for existing agents are directly inherited from DyLAN and detailed in the original paper Liu et al. (2024).

The prompt templates uniquely for OMAC include those designed for the Semantic Initializer and the Contrastive Comparator across the five optimization dimensions. Furthermore, as explained in Section 3.2, the only variation in the prompts for the two actors across different tasks lies in the contextual description of the MAS and the given task. Therefore, we present here the prompt templates for these two actors across the five optimization dimensions on general reasoning task. For details on other tasks, please refer to our code repository at: https://anonymous.4open.science/r/OMAC-Sub-3FF8.

Prompts of the Semantic Initializer for five dimensions are as follows:

Table 19: Prompts of Semantic Initializer for five dimensions.

| | |
|---|---|
| **Fun-1** | Generate `{initialization number}` distinct prompts to instruct an LLM to resolve some general reasoning problems acting as the given role: `{optimized agent role}`.
Each prompt should guide the model to accurately and efficiently resolve problems while adhering to the specified role.
Each prompt must begin strictly with the following content: `{basic description of the optimized agent}`. Then, you should consider adding more detailed, logical, and through instructions, which can help the LLM resolve problems better acting as the given role.
Do not output anything currently. Instead, I will provide a sequence number, and you should return only the corresponding prompt one by one.
Do not create any specific instances of the problems in the prompt, cause they are not provided now.
Ensure that the generated prompts follow the given example format but differ in content and structure from the example itself. The example is as follows:

`{one-shot example}`. |
| **Fun-2** | Generate `{initialization number}` distinct prompts to instruct an LLM to resolve some general reasoning problems related to math, hard science, humanities, and social sciences. There are some existing agents in the system to resolve the problems, whose roles are: `{roles of existing agents}`.
You need to generate some new roles and prompts for the LLM to better resolve the problems.
First, determine the roles of these prompts. Next, create the prompts that instruct the LLM to resolve problems based on the defined roles.
Do not output all the generated roles and prompts at once. Instead, I will request either the k-th role or the k-th prompt individually. When asked, directly output the corresponding content of the role name or prompt one at a time.
Do not create any instances of the problems in the prompt, cause they are not provided now.
You can decide the content and detailed functional instructions of the roles and prompts. You may consider adding more detailed instructions to help the LLM resolve problems.
The following is an example of a role and the corresponding prompt (also ensure your output is different from the example role and prompt):

`{one-shot example}`. |

| | |
|---|---|
| **Str-1** | Generate {`initialization number`} distinct prompts for an LLM to choose some top agents best suited for resolving some general reasoning problems related to math, hard science, humanities, social sciences, etc.
Don't directly output all the generated prompts. I will provide you the sequence number of the prompt. Then you should directly output the content text of the corresponding prompt one by one.
Each prompt should decide and specify the number of the chosen agents. The minimal number is 4 and maximum number is 7.
Each prompt should help to accurately and efficiently identify the top agents best suited for problem-solving.
Note that all information about the task and candidate agents has been previously provided as the context. The prompt generated here will be added to the context to form the final prompt for agent selection.
You may consider adding more detailed and thorough instructions to help the LLM select the top agents better.
The following is an example of a prompt (also ensure your output is different from the example prompt):

{`one-shot example`}. |
| **Str-2** | Generate {`initialization number`} distinct prompts for an LLM to choose some top solutions for best resolving some general reasoning problems related to math, hard science, humanities, social sciences, etc.
Don't directly output all the generated prompts. I will provide you with the sequence number of the prompt. Then you should directly output the content text of the corresponding prompt one by one.
You can decide the number of the chosen solutions and the content of the prompt. The number of solutions should be between 2 and 7.
The prompt should help to accurately and efficiently select the top solutions that resolve the given problems best.
Note that all the solutions and the problem have been previously provided as the context. The prompt generated here will be added to the context to form the final prompt for solution selection.
You may consider adding more detailed and thorough instructions to help the LLM select the top solutions better.
The generated prompt should specify the output format like the given example (also ensure that it is different from the example prompt):

{`one-shot example`}. |
| **Str-3** | Generate {`initialization number`} distinct prompts for an LLM to choose some top candidate agents whose generated solutions to some general reasoning problems may be useful as inputs for the current agent to produce improved solutions.
Don't directly output all the generated prompts. I will provide you with the sequence number of the prompt. Then you should directly output the content text of the corresponding prompt one by one.
You should decide the number of chosen agents and the content of the prompt. The number of chosen agents should be between 4 and 7.
Each prompt should help to accurately and efficiently identify the top candidate agents whose generated solutions are helpful to be taken as input for the current agent.
Note that all information about the candidate agents and the current agent has been previously provided as the context. The prompt generated here will be added to the context to form the final prompt for agent selection.
You may consider adding more detailed and thorough instructions to help the LLM select the candidate agents better.
The following is an example of a prompt (also ensure your output is different from the example prompt):

{`one-shot example`}. |

Prompts of the Contrastive Comparator for five dimensions are as follows:

Table 20: Prompts of Contrastive Comparator for five dimensions.

| | |
|---|---|
| **Fun-1** | Generate and output a child prompt for an LLM to resolve some general reasoning problems acting like the given role: {optimized agent role}. At the end, a pair of parent prompts is provided: one positive and one negative. The positive parent prompt has been shown to be more effective and efficient in guiding the LLM to resolve problems following the given role. Your task is to carefully compare the two parent prompts, identifying the key reasons why the positive parent prompt performs better. Based on these insights, generate and output a child prompt that further improves upon the positive parent prompt to enhance problem-solving. Do not create any instances of the problem in the prompt, cause they are not provided now. The child prompt must begin strictly with the following content: {basic description of the optimized agent}. Then, you can consider adding more detailed, logical, and through instructions based on the insights you have gained from the comparison. Output only the content of the child prompt excluding the reasoning process. Here is the positive-negative pair of parent prompts: {positive/negative prompts}. |
| **Fun-2** | Generate and output a pair consisting of a role name and its corresponding prompt, designed to resolve some general reasoning problems (related to math, hard science, humanities, social sciences, etc.). First, determine the role of the LLM. Next, create a prompt that effectively instructs the LLM to resolve problems based on this role. I will provide two parent role-prompt pairs: one positive and one negative. The positive pair has been proven to be more effective in guiding the LLM to generate high-quality solutions for general problems. Your task is to carefully analyze both parent pairs, identifying the factors that make the positive pair superior. Based on this analysis, generate and output a child role and prompt pair that improves upon the positive parent pair and leads to even better problem resolution. The child prompt must be distinct from both parent prompts while incorporating the lessons learned from their comparison. Do not output the role and prompt immediately. I will request them separately, and when asked, provide only the corresponding content—either the role name or the prompt. Here is the positive-negative pair of parent prompts: {positive/negative prompts}. |
| **Str-1** | Create and output a child prompt for an LLM to choose some top agents that best suited for resolving some general reasoning problems related to math, hard science, humanities, social sciences, etc. I will provide you with a pair of parent prompts. Then you should only output a child prompt according the following instructions: The positive parent prompt is proven to be more helpful and efficient to instruct the LLM to select more useful and effective agents for problem resolution. You should carefully compare the two parent prompts, finding the potential reasons why the positive parent prompt is better than the negative parent prompt. Based on that, you should generate and output a child prompt that can help to choose top agents more effectively and efficiently than the positive prompt. The child prompt should follow the format of the parent prompts. The child prompt should be different from the parent prompts. And directly output the content text of the child prompt. Here is the positive-negative pair of parent prompts: {positive/negative prompts}. |

| | |
|---|---|
| **Str-2** | Create and output a child prompt for an LLM to choose some top solutions for resolving some general reasoning problems (related to math, hard science, humanities, social sciences, etc.) best. 
 I will provide you with a pair of parent prompts. Then you should only output a child prompt according the following instructions: 
 The positive parent prompt is proven to be more helpful and efficient to instruct the LLM to select more useful and effective solutions to resolve the problems. 
 You should carefully compare the two parent prompts, finding the potential reasons why the positive parent prompt is better than the negative parent prompt. Based on that, you should generate and output a child prompt that can help to choose top solutions more effectively and efficiently than the positive prompt. 
 The child prompt should follow the format of the parent prompts. 
 The child prompt should be different from the parent prompts. And directly output the content text of the child prompt. 
 Here is the positive-negative pair of parent prompts: `{positive/negative prompts}`. |
| **Str-3** | Create and output a child prompt for an LLM to choose some candidate agents whose generated solutions to some general reasoning problems may be useful as inputs for the current agent to produce improved solutions. 
 I will provide you a pair of parent prompts. Then you should only output a child prompt according the following instructions: 
 The positive parent prompt is proven to be more helpful and efficient to instruct the LLM to select more useful and effective agents. 
 You should carefully compare the two parent prompts, finding the potential reasons why the positive parent prompt is better than the negative parent prompt. Based on that, you should generate and output a child prompt that can help to choose top agents more effectively and efficiently than the positive prompt. 
 The child prompt should follow the format of the parent prompts. 
 The child prompt should be different from the parent prompts. And directly output the content text of the child prompt. 
 Here is the positive-negative pair of parent prompts: `{positive/negative prompts}`. |

## D.2 Examples

Besides the prompts for the two actors, we further provide examples of instruction prompts before and after optimizing corresponding to the five optimization dimensions on general reasoning task.

| Dimension | Fun-1 |
|---|---|
| **Default Prompt** | You are a mathematician. You are good at math games, arithmetic calculation, and long-term planning. Please provide a step-by-step approach to solve some provided mathematical problems. |
| **Optimized Prompt** | You are a mathematician. You are good at math games, arithmetic calculation, and long-term planning. 
 When devising a step-by-step strategy for solving a complex optimization problem involving multiple variables and constraints, consider breaking down the problem into smaller, more manageable sub-problems. 
 Analyze the objective function and constraints to understand their impact on the feasible solution space. Explore various solution methods such as linear programming, gradient descent, or integer programming to determine the most suitable approach for the problem at hand. 
 Finally, implement the chosen method meticulously while constantly evaluating its efficiency and accuracy. If necessary, iterate on the strategy to improve the optimization process. 
 Explain the rationale behind each step of your chosen approach. |

| Dimension | Fun-2 |
|---|---|
| **Default Prompt** | You are a super-intelligent AI assistant capable of performing tasks more effectively than humans. |
| **Optimized Prompt** | You are a Data Analyst. You need to provide a detailed explanation of how to use statistical techniques to identify and analyze patterns for the data in the given question.
Provide a step-by-step guide on how to conduct the analysis for a given dataset. Also, provide a clear and detailed explanation of the process for selecting an appropriate statistical approach. |

| Dimension | Str-1 |
|---|---|
| **Default Prompt** | Here is the task and question: {task context}.
These are the agents and their functional description: {candidate agents' functionalities}.

Take functionality, efficiency, and necessity into consideration, choose top 5 agents best suited for resolving the given problem. Think it step by step. Put your answer in the form like [1,3,4,5,6] at the end of your response. |
| **Optimized Prompt** | Here is the task and question: {task context}.
These are the agents and their functional description: {candidate agents' functionalities}.

To address general reasoning problems across various disciplines such as math, hard science, humanities, and social sciences, it is crucial to identify the top 6 agents with exceptional problem-solving abilities and expertise in diverse areas.
These agents should demonstrate proficiency in critical thinking, logical reasoning, and analytical skills to effectively resolve multifaceted problems.
Evaluate the candidates based on their demonstrated knowledge, adaptability, and capability in tackling complex reasoning challenges.
After carefully assessing these criteria, provide your response in the form [1,2,3,4,5,6] at the end of your submission. |

| Dimension | Str-2 |
|---|---|
| **Default Prompt** | Here is the task and question: {task context}.
These are the solutions to the problem from other agents: {previous agents' solutions}.

Please choose the best 2 solutions and think step by step. Put your answer in the form like [1,2] or [3,4] at the end of your response. |
| **Optimized Prompt** | Here is the task and question: {task context}.
These are the solutions to the problem from other agents: {previous agents' solutions}.

Analyze the given context thoroughly and choose the top 3 solutions based on their ability to accurately and efficiently resolve the given problems. The selected top solutions should be effective in resolving reasoning problems in various fields including math, science, humanities, and social sciences.
Consider practical applicability, logical soundness, and clarity of each solution. Then think step by step to clearly explain how each solution can be applied in different scenarios.
Please put your answer in the form like [1,2,3] at the end of your response. |

| Dimension | Str-3 |
| --- | --- |
| **Default Prompt** | Here is the functional description of the current agent: {`current agent' functionality`}.
These are the candidate agents and their functional description: {`candidate agents' functionalities`}.

Take functionality, efficiency, and necessity into consideration. Select the top 5 candidate agents whose generated solutions to some general reasoning problems can be mostly useful as inputs for the current agent to produce improved solutions. Think it step by step.
Put your answer in the form like [1,2,3,4,5] at the end of your response. |
| **Optimized Prompt** | Here is the functional description of the current agent: {`current agent's functionality`}.
These are the candidate agents and their functional description: {`candidate agents' functionalities`}.

Consider the agents whose generated solutions are most likely to improve the current agent's problem-solving capabilities.
Select the top 4 candidate agents based on the effectiveness, practicality, and relevance of their solutions.
Consider their ability to address complex challenges, think outside the box, and produce innovative perspectives that could benefit the current agent in enhancing its problem-solving capabilities.
Prioritize agents whose solutions offer a fresh approach, logical reasoning, and effective problem-solving strategies.
Present your answer in the format [1,2,3,4] at the end of your response. |

# E   LLM USAGE

Throughout the writing of this paper, LLMs were used exclusively for improving clarity of expression and correcting typographical or grammatical errors. No other substantive assistance was employed.

