# OpenReview forum: "OMAC: A Broad Optimization Framework for LLM-Based Multi-Agent Collaboration"
_ICLR.cc/2026/Conference — Submitted to ICLR 2026_

### Official Review · Reviewer_KmRj · 2025-10-15

**Soundness:** 3
**Presentation:** 3
**Contribution:** 2
**Rating:** 2
**Confidence:** 5

**Summary:**

This paper introduces OMAC, a general framework for systematically optimizing LLM-based Multi-Agent Systems (MAS). The authors identify five key optimization dimensions: two related to agent functionality (optimizing existing agents, constructing new ones) and three related to collaboration structure (candidate selection, dynamic participation, communication flows). The core of OMAC consists of two LLM-powered actors: a "Semantic Initializer" that generates an initial diverse set of agents or controllers, and a "Contrastive Comparator" that iteratively refines them by reasoning over high-performing (positive) and low-performing (negative) pairs sampled based on performance on a training set. The framework also supports joint, iterative optimization across multiple dimensions. Extensive experiments on code generation, general reasoning, and arithmetic reasoning tasks demonstrate that OMAC significantly outperforms existing baselines, with further gains achieved through multi-dimensional optimization.

**Strengths:**

1. The paper is well-written.

2. The motivation of the paper is clear.

3. Comprehensive and Systematic Framework: The paper proposes a holistic and principled framework for MAS optimization that is a significant step beyond ad-hoc, handcrafted design.

**Weaknesses:**

1. Limited Novelty of Core Components: The core ideas behind the Semantic Initializer and Contrastive Comparator are conceptually similar to principles already established in the literature on prompt and agent optimization [1,2,8,9,10]

2. Insufficient Comparison with Latest Baselines: The experimental evaluation is missing comparisons against several recent and highly relevant baselines in multi-agent system optimization, such as Aflow [1], AgentSquare [2], MaAS [3], and G-designer [4]. The paper includes a comparison with Aflow, but it is limited to the MBPP benchmark and is placed in the appendix (Appendix B.2.5, Table 11) rather than being integrated into the main experimental results across all benchmarks (Section 4.1, Tables 1-3).

3. Use of Potentially Saturated Benchmarks: The currently used benchmark is somewhat outdated. The most advanced models or methods have already achieved very high scores. To more robustly demonstrate the advantages of the OMAC framework, evaluation on more recent and challenging benchmarks such as GAIA [5], SWE-bench [6], or HLE [7] would be more convincing and provide a better measure of its capabilities on complex, unsolved problems.

4. Lack of Evaluation on Open-Source Models: The paper's empirical evidence relies exclusively on proprietary GPT-series models. The experiments should be conducted on a diverse range of open-source LLMs.

5. Incomplete Computational Cost Analysis: A more thorough analysis would involve a direct comparison of the training cost against other MAS optimization frameworks. Visualizing this trade-off, for instance with a Pareto front plotting performance against computational cost, would provide a much clearer picture of OMAC's efficiency relative to its competitors.

6. Outdated Related Work Section: The Related Work section (Section 5) should be updated to include a discussion of the latest relevant works.

Reference

[1]Zhang J, Xiang J, Yu Z, et al. AFlow: Automating Agentic Workflow Generation[C]//The Thirteenth International Conference on Learning Representations.

[2]Shang Y, Li Y, Zhao K, et al. AgentSquare: Automatic LLM Agent Search in Modular Design Space[C]//The Thirteenth International Conference on Learning Representations.

[3]Zhang G, Niu L, Fang J, et al. Multi-agent Architecture Search via Agentic Supernet[C]//Forty-second International Conference on Machine Learning.

[4]Zhang G, Yue Y, Sun X, et al. G-Designer: Architecting Multi-agent Communication Topologies via Graph Neural Networks[C]//Forty-second International Conference on Machine Learning.

[5]Mialon G, Fourrier C, Wolf T, et al. Gaia: a benchmark for general ai assistants[C]//The Twelfth International Conference on Learning Representations. 2023.

[6]Jimenez C E, Yang J, Wettig A, et al. SWE-bench: Can Language Models Resolve Real-world Github Issues?[C]//The Twelfth International Conference on Learning Representations.

[7]Phan L, Gatti A, Han Z, et al. Humanity's last exam[J]. arXiv preprint arXiv:2501.14249, 2025.

[8]Hu S, Lu C, Clune J. Automated Design of Agentic Systems[C]//The Thirteenth International Conference on Learning Representations.

[9]Yuksekgonul, M., Bianchi, F., Boen, J. et al. Optimizing generative AI by backpropagating language model feedback. Nature 639, 609–616 (2025). https://doi.org/10.1038/s41586-025-08661-4

[10]Shinn N, Cassano F, Gopinath A, et al. Reflexion: Language agents with verbal reinforcement learning[J]. Advances in Neural Information Processing Systems, 2023, 36: 8634-8652.

**Questions:**

Please address the issue I raised in weaknesses.

---

> ### Author Response · Authors · 2025-11-21
>
> Dear Reviewer KmRj,
>
> Thanks for your positive feedback on the motivation, significance, soundness, and methodology of our work.
> Your thoughtful and constructive comments will help us to further clarify our methods and innovations. We hope that our response below can be helpful to address your concerns.
>
> **W1. Novelty**
>
> Thank you for suggesting these relevant works. We'd like to clarify that our claim of novelty does not lie in the creation of the individual optimization components themselves. As noted by the reviewer, similar ideas are indeed principled and straightforward in the context of LLM optimization. Rather, as summarized at the end of the Introduction section, the novelty of our work lies in the conceptual framing and integration of these components within a unified optimization framework for MAS optimization.
>
> Specifically, we summarize our contributions as fourfold:
>
> * We identify five key optimization dimensions that comprehensively cover both agent functionality and collaboration structure in MAS.
>
> * We propose a general optimization framework that **utilizes** these two actors to optimize each of these five dimensions.
>
> * We further introduce an iterative algorithm that enables joint optimization across multiple dimensions.
>
> * We empirically validate our framework on multiple standard benchmarks against several MAS baselines, demonstrating consistent performance improvements and practical value.
>
> Therefore, while OMAC leverages existing principled ideas and notations in certain components, we believe the conceptual generalization, unified optimization framework, and empirical validation together constitute sufficient novelty and contribution to the field.
>
> **W2. Baselines**
>
> We thank the reviewer for suggesting these relevant works. First, we'd like to clarify that our experiments already include several recent and highly relevant baseline methods, such as DyLAN (COLM 2024) [1], AgentVerse (ICLR 2024) [2], and ADAS (ICLR 2025) [3]. In particular, DyLAN and ADAS represent some of the recent and well-cited approaches in multi-agent system optimization research. Additionally, our previous version already included comparisons with AFlow (ICLR 2025) [4] on the more challenging MBPP benchmark (Appendix B.2.5).
>
> Nevertheless, we do agree that extending the comparison with AFlow across other benchmarks would further strengthen the comprehensiveness of our evaluation. Accordingly, with additional computational resources and time, we have expanded our experiments to include these comparisons in the updated manuscript (Table 1-3). As shown in both the revised paper and the accompanying table below, by comparing with the latest MAC optimization methods like **ADAS and AFlow that were published in 2025**,
> OMAC consistently outperforms these SOTA MAS optimization methods, further validating the effectiveness and generality of our proposed framework.
>
> |       |ADAS |AFLOW|OMAC   |       |       |       |       |       |       |     |     |     |     |
> |-------|-----|-----|-------|-------|-------|-------|-------|-------|-------|-----|-----|-----|-----|
> |       |     |     |Fun-1.1|Fun-1.2|Fun-1.3|Fun-1.4|Fun-1.5|Fun-1.6|Fun-1.7|Fun-2|Str-1|Str-2|Str-3|
> | Pass@1|75.61|85.63|88.39  |86.31  |88.87  |89.25  |88.74  |88.39  |88.34  |86.77|86.76|86.92|87.55|
>
> **W3. Benchmarks**
>
> Thank you for suggesting these benchmarks. First, we wish to emphasize that the benchmarks used in our main paper (HumanEval, MMLU, and MATH) are well-established and widely adopted in recent MAS research. Prominent studies such as DyLAN (COLM 2024) [1], AgentVerse (ICLR 2024) [2], ADAS (ICLR 2025) [3], AFlow (ICLR 2025) [4], and LLM Debate (ICML 2023) [5] have all utilized these benchmarks to evaluate multi-turn MAS and/or their optimization methods. We've also included results on the more challenging MBPP benchamrk in Appendix B.2.5. Therefore, we consider our chosen benchmarks to be representative, reliable, and directly comparable with existing literature.
>
> Nevertheless, we agree that incorporating additional recent and challenging benchmarks can provide further evidence of OMAC’s capability in solving complex tasks. Following the reviewer’s suggestion, we have now included the GAIA benchmark for comparison and have presented the corresponding results and discussion in **Appendix B.2.5**. As shown in both the revised manuscript and the accompanying table below, OMAC continues to demonstrate substantial performance improvements over state-of-the-art MAS optimization approaches such as ADAS, DyLAN, and AFlow.
>
>
> |       |ADAS |AFLOW|DyLAN|OMAC   |       |       |       |       |       |       |     |     |     |     |
> |-------|-----|-----|-----|-------|-------|-------|-------|-------|-------|-------|-----|-----|-----|-----|
> |       |     |     |     |Fun-1.1|Fun-1.2|Fun-1.3|Fun-1.4|Fun-1.5|Fun-1.6|Fun-1.7|Fun-2|Str-1|Str-2|Str-3|
> | Pass@1|3.87 |8.23 |9.82 |12.14  |10.92  |11.73  |13.04  |11.97  |13.74  |10.70  |14.24|11.93|10.16|12.88|

---

> ### Author Response · Authors · 2025-11-21
>
> **W4. Open-Source Models**
>
>
>
> Thank you for the helpful suggestion. In our original experiments, we primarily followed one of our main baselines, DyLAN, and adopted GPT-3.5-turbo and GPT-4o-mini as backbone LLMs to ensure a direct and fair comparison. However, we fully agree that incorporating open-source LLMs can further demonstrate the generality and robustness of OMAC.
>
> Accordingly, we have conducted additional experiments using two open-source models, DeepSeek-V2.5 and Qwen-2.5-72B, as backbones. The corresponding results have been added to Appendix B.2.4 in the updated manuscript. As shown in the revised paper, OMAC consistently outperforms all baseline methods, confirming its strong generalization ability across both proprietary and open-source LLMs.
>
>
>
> **W5. Cost Analysis**
>
>
>
> Thank you for the valuable suggestion regarding a more comprehensive computational cost analysis. While we have previously included comparisons of both inference and training costs in Appendix B.2.6, we agree that a more direct and detailed comparison against other MAS optimization methods would further highlight OMAC’s efficiency.
>
> Following the reviewer’s suggestion, we have extended Appendix B.2.6 in the updated manuscript to include a comprehensive comparison among AgentVerse, DyLAN, AFlow, and OMAC in terms of training cost, inference cost, and overall performance (see Table 17). The results show that OMAC achieves substantial improvements in both inference efficiency and performance than all compared baselines, with only a minor increase in training cost compared to DyLAN. As demonstrated both in the manuscript and the accompanying table below,  these findings further confirm OMAC’s strong cost–performance balance and practical applicability for real-world multi-agent systems.
>
>
> |                   |AgentVerse |DyLAN |AFlow  |  OMAC (avg.) |
> |-------------------|-----------|------|-------|--------------|
> |Training Cost (\$) |-          |4.62  |8.24   |         5.15 |
> |Inference Cost (\$)|0.1928     |0.1288|0.1134 |        0.0974|
> |Accuracy (\%)      |78.26      |85.74 |85.63  |        87.68 |
>
>
>
> **W6. Related Work Section**
>
>
>
> We appreciate the reviewer’s suggestion to expand the Related Work section to include a broader discussion of recent and relevant studies. With increased page limit of the main paper, we have now extended the Related Work section (Section 5) of the updated manuscript. This expanded section provides a more comprehensive overview and comparison with the most recent MAS optimization methods.  We kindly refer the reviewer to this updated content for the detailed discussion.
>
>
> ---
>
> References
>
> [1] Liu, Zijun, et al. "A dynamic llm-powered agent network for task-oriented agent collaboration." First Conference on Language Modeling. 2024.
>
> [2] Chen, Weize, et al. "Agentverse: Facilitating multi-agent collaboration and exploring emergent behaviors in agents." ICLR. 2024.
>
> [3] Hu, Shengran, Cong Lu, and Jeff Clune. "Automated design of agentic systems." ICLR. 2025.
>
> [4] Zhang, Jiayi, et al. "Aflow: Automating agentic workflow generation." ICLR. 2025.
>
> [5] Du, Yilun, et al. "Improving factuality and reasoning in language models through multiagent debate." ICML. 2023.

---

> > ### Comment · Reviewer_KmRj · 2025-11-28
> >
> > I thank the authors for their detailed response and appreciate the significant effort put into the rebuttal, particularly the additional experiments involving open-source models, the inclusion of the GAIA benchmark, and the cost analysis. These revisions have addressed some of my initial concerns.
> >
> > However, regarding the empirical evaluation of benchmarks and baselines, I remain unconvinced that the current results are sufficient to fully validate the framework's superiority. My concerns are as follows:
> >
> > 1.  **Limited Comparisons:** The comparison with the latest SOTA baselines (such as MaAS) still appears limited in scope. I believe more extensive empirical evidence is necessary to robustly demonstrate the advantages of OMAC.
> > 2.  **Marginal Improvements:** Regarding the cost analysis and performance trade-offs, the reported performance gains appear marginal.
> >
> > Given these remaining issues, while I acknowledge the improvements made to the manuscript, I believe the paper requires further solid empirical verification to meet the bar for acceptance. Consequently, I am raising my score to **4**.

---

> > > ### Author Response · Authors · 2025-12-02
> > >
> > > We sincerely thank the reviewer for their continued engagement and thoughtful follow-up comments. We are glad that our additional experiments and clarifications have addressed the reviewer’s concerns regarding novelty, experiments on open-source models, the inclusion of recent benchmarks, and cost analysis. To further address the remaining questions on baseline comparisons and performance improvements, we provide additional clarification below.
> > >
> > > First, regarding the requested comparison with MaAS [1], we'd like to clarify that MaAS is an approach that **exclusively focuses on structural optimization** of MAS, as illustrated in Section 5. In contrast, OMAC is designed to optimize both the functional and structural dimensions of MAS. In the MaAS implementation, the authors directly adopt well-optimized and near-optimal agent configurations from prior work like AgentCoder [2] and ReAct [3] as fixed functional components, and only optimize the collaboration structure. OMAC, however, begins with simple and minimally engineered agent configurations and is designed to optimize both agent functionality and MAS structure. Because MaAS relies on externally optimized functional agents and does not address functional optimization itself, we believe it is not directly comparable to OMAC in terms of scope or methodology.
> > > Instead, we have included ADAS and AFlow, two of the most recent SOTA MAS optimization **approaches (ICLR 2025) that target both functional and structural optimization**. As such, we believe that our experiments already include appropriate, representative, and up-to-date SOTA baselines for a fair and meaningful comparison.
> > >
> > >
> > > Second, regarding the magnitude of performance improvement, we'd like to clarify that OMAC’s gains should not be considered marginal. For single-dimension optimization, **OMAC achieves average relative improvements of 3.6\%, 2.8\%, and 1.8\%** per dimension on the MATH, HumanEval, and MBPP benchmarks, surpassing strong and recent baselines such as DyLAN (COLM 2024), ADAS (ICLR 2025), and AFlow (ICLR 2025).
> > > More importantly, OMAC’s ability to jointly **optimize multiple dimensions leads to substantially larger improvements** than optimizing individual dimensions in isolation. For example, as shown in Figure 4, jointly optimizing Fun-1.1 and Fun-1.2 increases the improvement from 2.9\% (single-dimension) to 9.6\%. Similar patterns are consistently observed in Figures 5–6.
> > > For comparison, the method suggested by the reviewer, **MaAS, achieves only 1.0\%, 2.1\%, and 0.6\% improvement** over its strongest baselines on the same three benchmarks according to the original paper.
> > > In summary, these results demonstrate that OMAC delivers systematic, meaningful, and compounding improvements in MAS optimization rather than incremental ones.
> > >
> > > ---
> > >
> > > References
> > >
> > > [1] Zhang, Guibin, et al. "Multi-agent architecture search via agentic supernet." ICLR. 2025.
> > >
> > > [2] Huang, Dong, et al. "Agentcoder: Multi-agent-based code generation with iterative testing and optimisation." arXiv preprint arXiv:2312.13010 (2023).
> > >
> > > [3] Yao, Shunyu, et al. "React: Synergizing reasoning and acting in language models." ICLR. 2023.

---

### Official Review · Reviewer_yLHQ · 2025-10-20

**Soundness:** 3
**Presentation:** 4
**Contribution:** 2
**Rating:** 6
**Confidence:** 4

**Summary:**

The paper proposes OMAC, a general supervised optimization framework for LLM-based multi-agent systems (MAS) engaged in multi-step collaboration. It identifies five key optimization dimensions—two functional (optimizing existing agents, constructing new agents) and three structural (candidate selection, dynamic participation, communication flow)—and introduces a unified algorithm using two LLM-powered actors: the Semantic Initializer (for diverse prompt generation) and the Contrastive Comparator (for iterative refinement via performance-guided contrastive reasoning). Experiments on code generation, general reasoning, and arithmetic reasoning show consistent improvements over strong baselines, with further gains from joint multi-dimension optimization.

**Strengths:**

1) OMAC offers a systematic decomposition of MAS optimization into five well-motivated dimensions, going beyond prior work (e.g., DyLAN) that focuses only on team composition. The use of contrastive reasoning guided by supervised performance signals is an approach to prompt/controller optimization in MAS.

2) The paper is well-structured and clearly written. Key concepts (e.g., dimensions, actors, workflow) are introduced intuitively. The optimization process is easy to follow despite its generality.

**Weaknesses:**

1) The proposed framework is essentially a form of rejection fine-tuning (RFT), which samples positive examples to train the LLMs. However, a large body of work in agentic reinforcement learning has already explored optimizing agents or managing prompts via supervised fine-tuning (SFT), rejection fine-tuning (RFT), reinforcement learning (RL), or prompt evolution [1–3]. The authors should compare their approach with these existing methods and clearly articulate the key differences.

2) LLM generation–based agent construction also does not rely on human prior knowledge. In this context, why is optimizing agent team composition using supervised signals preferable to unsupervised approaches?

3) The rationale for optimizing these five specific dimensions—each of which appears largely independent—is not sufficiently justified. Optimizing across five separate dimensions reads more like a collection of incremental improvements rather than a cohesive, unified contribution, which undermines the overall novelty of the paper.

4) OMAC optimizes prompts at test time using ground-truth labels, which constitutes an unfair comparison with ByLAN. The authors should include comparisons against supervised methods that also leverage test-time information, such as test-time training approaches.

5) There are several grammatical and typographical issues. For example:
   Original: “in handling tasks within sophisticated environments Li et al. (2023a); Wang et al. (2023b).”
   Suggested revision: “in handling tasks within sophisticated environments (Li et al., 2023a; Wang et al., 2023b).”

References
[1] Wong, M., Ong, Y. S., Gupta, A., et al. Prompt evolution for generative AI: A classifier-guided approach. In *2023 IEEE Conference on Artificial Intelligence (CAI)* (pp. 226–229). IEEE, 2023.
[2] Agrawal, L. A., Tan, S., Soylu, D., et al. GEPA: Reflective prompt evolution can outperform reinforcement learning. *arXiv preprint arXiv:2507.19457*, 2025.
[3] Guo, W., Yang, J., Yang, K., Li, X., Rao, Z., Xu, Y., & Niu, D. (2024). Instruction Fusion: Advancing Prompt Evolution through Hybridization. In *Proceedings of the 62nd Annual Meeting of the Association for Computational Linguistics (Volume 1: Long Papers)* (pp. 3883–3893). Association for Computational Linguistics.

**Questions:**

1) Can the proposed model optimize all five dimensions simultaneously? Why does the framework train them iteratively?

2) How does the proposed method compare with approaches that optimize agents via test-time training or prompt evolution? The authors should include more trainable baselines for a fair comparison.

3) How do supervised methods perform on the same tasks? A comparison with such methods would help contextualize the claimed advantages of the proposed approach.

4) Moreover, the method requires full MAS evaluation on the training set per candidate, which is expensive. The author should compare the complexity with existing models.

---

> ### Author Response · Authors · 2025-11-21
>
> Dear Reviewer yLHQ,
>
>
> Thanks for your positive feedback on the motivation, significance, soundness, and quality of our work.
> Your thoughtful and constructive comments will help us to further clarify our methods and innovations. We hope that our response below can be helpful to address your concerns.
>
>
> **W1. Discussion of Other Approaches**
>
>
> Thank you for raising this valuable point. We'd like to clarify that while our approach shares certain intuitions with rejection fine-tuning, it differs significantly in both motivation and mechanism. As described in Section 3.3, OMAC leverages the LLM’s contrastive reasoning ability during optimization. Specifically, we construct positive–negative sample pairs and feed them into a Contrastive Comparator, which is instructed to reason about their differences. The comparator then refines the targeted optimization dimension by amplifying factors associated with positive outcomes and suppressing those linked to negative ones. In contrast, rejection fine-tuning typically relies on filtering or selecting positive examples for fine-tuning, without explicitly engaging the model’s contrastive reasoning process.
>
> Regarding prompt evolution, one of our baselines, ADAS (ICLR 2025) [1], can be viewed as an instance of this approach applied to MAS optimization. For gradient-based methods such as supervised fine-tuning and RL-based optimization, the existing literature on our studied problem setting, which is **multi-step collaboration among multiple LLM-based agents, remains very limited** to the best of our knowledge.
>
> Nevertheless, we fully agree that a broader discussion of related optimization paradigms can provide additional clarity, context, and insight for future works. Following the reviewer’s suggestion, we have expanded the related work section (Section 5) to include a more comprehensive comparison and discussion of these alternative approaches. Also, we've included a Future Work section in Appendix C.2 to discuss future research on combining OMAC with these directions. We respectfully refer the reviewer to these sections for the detailed discussion.
>
>
>
> **W2. Agent Team Optimization**
>
>
> As described in Section 3.1 and Appendix A, OMAC optimizes LLM-based controllers to determine the composition of the agent team. If OMAC is not applied and the default prompt alone is used to instruct this controller, the resulting setup corresponds to the unsupervised case mentioned by the reviewer. As detailed in Appendix A, the intuition and benefit of incorporating supervised signals lie in optimizing the controller to selectively include only the agents most beneficial for the given task, thereby reducing potential interference and improving both the effectiveness and efficiency of the multi-agent system. Through contrastive reasoning guided by supervised signals, the LLM-based controller is optimized to make more accurate and context-sensitive decisions about team composition, based on the given context of all available agents and the current task to resolve.
>
> The principle of agent construction optimization is the same. While new agents can indeed be generated directly via prompting the LLM, **OMAC further refines the agent construction by using supervised signals** to evaluate and contrast the quality of different constructed agents. This enables contrastive refinement, guiding the system toward generating agents that contribute more effectively to collaborative performance.
>
>
>
> **W3. Optimization Rationale**
>
>
>
> Thanks for raising this valuable point. We'd like to clarify that OMAC introduces two algorithms designed to optimize MAS dimensions, one for single-dimension optimization and another for multi-dimension optimization. As described in Appendix A, the five optimization dimensions are derived by conceptualizing the multi-agent collaboration process as an information flow through a graph, where agents correspond to nodes and communication paths to edges. Because each dimension represents a distinct aspect of this graph-based formulation, the five dimensions are conceptually independent. Consequently, OMAC’s ability to optimize each dimension individually as well as optimize multiple dimensions jointly is considered a deliberate and advantageous design choice. This flexibility allows OMAC to target specific aspects of MAS functionality or structure when needed, while also enabling coordinated optimization across dimensions for compounded improvements.
>
> It is also worth noting that this modular optimization approach is consistent with existing MAS optimization methods such as DyLAN [2], ADAS [1], and AFlow [3], which likewise optimize different characteristics of MAS using distinct operators or separate optimization paradigms. OMAC builds upon this principle but generalizes it under a unified and extensible framework.

---

> ### Author Response · Authors · 2025-11-21
>
> **W4. Supervised Methods**
>
>
>
> First, we'd like to clarify that OMAC **does not use ground-truth labels at test time**. As detailed in Section 3.2 and Section 3.4, all supervised optimization is performed using ground-truth labels only on the training set, which is strictly disjoint from the test set. After the training phase, OMAC selects the agent and/or controller configurations that achieve the highest performance on the training data. These optimized components are then directly deployed within the MAS for inference and evaluation on the test set, without any access to ground-truth labels.
>
> To the best of our knowledge, existing research on supervised methods for MAS optimization in multi-step collaboration settings remains limited. Nonetheless, we agree that incorporating such baselines can be beneficial for completeness. Actually, **we've already included a comparison with a recent supervised work**, AFlow (ICLR 2025) [3], on the MBPP benchmark (Appendix B.2.5), which also leverages supervised signals of MAC optimization. However, AFlow uses them to guide Monte Carlo Tree Search for discovering effective agentic workflows, whereas OMAC employs supervised signals directly for contrastive reasoning-based MAS optimization, representing a distinct optimization paradigm.
>
> With the availability of additional computational resources, we have extended our comparison with AFlow to other benchmarks. As shown in the updated manuscript and the accompanying table, OMAC consistently outperforms SOTA MAS optimization methods, demonstrating its strong effectiveness and general applicability. We've also included relevant discussions in the extended related work section (Section 5).
>
>
>
> |       |ADAS |AFLOW|OMAC   |       |       |       |       |       |       |     |     |     |     |
> |-------|-----|-----|-------|-------|-------|-------|-------|-------|-------|-----|-----|-----|-----|
> |       |     |     |Fun-1.1|Fun-1.2|Fun-1.3|Fun-1.4|Fun-1.5|Fun-1.6|Fun-1.7|Fun-2|Str-1|Str-2|Str-3|
> | Pass@1|75.61|85.63|88.39  |86.31  |88.87  |89.25  |88.74  |88.39  |88.34  |86.77|86.76|86.92|87.55|
>
>
>
>
> **W5. Grammatical Issues and Typos**
>
> Thank you very much for carefully identifying these grammatical issues. We've thoroughly reviewed the manuscript and corrected the typos in the updated version.
>
>
>
>
> **Q1. Multi-Dimension Optimization**
>
>
>
> As we illustrated in Section 3.3, the rationale behind our iterative optimization strategy for multiple dimensions is to **preserve the effectiveness of the LLM’s contrastive reasoning process**. Specifically, by constraining variations within each positive–negative sample pair to a single dimension at a time, we maintain consistency across all other factors. This design enables the LLM-based Contrastive Comparator to clearly identify the underlying causes of performance differences, avoiding the confounding effects that arise when multiple interacting variables change simultaneously.
>
> In our original submission, we **have conducted experiments to validate this design**, with results and analyses presented at the end of Appendix B.2.3. These results show that simultaneously optimizing multiple dimensions, which requires the Contrastive Comparator to reason over multiple varying factors at once, leads to notably smaller performance gains and higher variance compared to the iterative approach. This finding supports the rationale and the necessity of the proposed iterative optimization pattern in OMAC. Nevertheless, we've included a relevant discussion of exploring other multi-dimension optimization techniques within OMAC in the Future Works section (Appendix C.2).
>
>
>
>
> **Q2/Q3. Comparison with Other Baselines**
>
>
> As discussed in our response to **W2**, we have already included ADAS (ICLR 2025) [1], a recent prompt evolution–based baseline, for comparison across all benchmarks. Additionally, as illustrated in response to **W4**, we have incorporated AFlow (ICLR 2025) [3] as a supervised optimization baseline on the MBPP benchmark (Appendix B.2.5). In our updated manuscript, we have now extended the comparison with AFlow to additional benchmarks as well.
>
> Across all these recent and advanced MAS optimization methods, OMAC consistently achieves superior performance. These results further reinforce the effectiveness and generality of our proposed framework in optimizing MAS. And we've included relevant discussions in the extended Related Work section (Section 5).

---

> ### Author Response · Authors · 2025-11-21
>
> **Q4. Computation Cost**
>
>
>
> We'd like to clarify that we have **already provided a detailed analysis of both inference and training costs** in Appendix B.2.6. Notably, OMAC significantly reduces inference-time computational cost by dynamically optimizing the collaboration structure, thereby outperforming other multi-agent frameworks in efficiency. We deem this as an important advantage of OMAC, as inference efficiency is critical for the practical deployment of agentic systems in real-world industrial settings.
>
> For training cost, by adopting a sampling-based strategy, OMAC can optimize a single MAS dimension for the code generation task at a cost of only approximately $5, while achieving an average relative performance improvement of over 2.2% compared to the strongest baseline. This high cost-effectiveness makes OMAC  attractive for industrial adoption, where even performance gains of around 1% can yield substantial practical and commercial value. As demonstrated both in the manuscript and the accompanying table below,  we believe this practical cost-performance balance underscores the real-world applicability of our approach.
>
>
> |                   |AgentVerse |DyLAN |AFlow  |  OMAC (avg.) |
> |-------------------|-----------|------|-------|--------------|
> |Training Cost (\$) |-          |4.62  |8.24   |         5.15 |
> |Inference Cost (\$)|0.1928     |0.1288|0.1134 |        0.0974|
> |Accuracy (\%)      |78.26      |85.74 |85.63  |        87.68 |
>
> ---
>
> References
>
> [1] Hu, Shengran, Cong Lu, and Jeff Clune. "Automated design of agentic systems." ICLR. 2025.
>
> [2] Liu, Zijun, et al. "A dynamic llm-powered agent network for task-oriented agent collaboration." First Conference on Language Modeling. 2024.
>
> [3] Zhang, Jiayi, et al. "Aflow: Automating agentic workflow generation." ICLR. 2025.

---

### Official Review · Reviewer_YvM6 · 2025-10-28

**Soundness:** 2
**Presentation:** 3
**Contribution:** 2
**Rating:** 4
**Confidence:** 4

**Summary:**

This paper introduces OMAC for systematically designing and optimizing Multi-Agent Systems. The method helps mitigate the current reliance on handcrafted, labor-intensive methods in MAS development. OMAC proposes an automated approach to optimize collaborative structure, communication patterns, and underlying prompts to achieve enhanced performance across complex tasks.

**Strengths:**

The core concept of introducing a systematic optimization framework to move beyond "handcrafted methods" for MAS design is meaningful. The implicit reliance on LLMs to perform meta-optimization on the agent configuration is a promising application of LLM capabilities. The proposed method achieves superior performance to the compared baselines.

**Weaknesses:**

1. The term "Broad Optimization" in the title is misleading, as the framework appears narrowly focused on optimizing communication/collaboration prompts. The paper does not demonstrate the capability to optimize critical non-prompt-based technical elements, such as the agent's action space (tool/API use), internal state representation (memory management and retrieval). This significantly limits the framework's claim to "broad" generality.

2. The underlying principle of optimization effectiveness remains unclear. Why only these five dimensions are considered? Are they manually defined? The authors are encouraged to provide more direct evidence why and how OMAC selects the optimal configuration.

3. Assuming the performance gain is marginal (e.g., around 1% in many tasks), this fails to justify the significant computational overhead and technical complexity introduced by the OMAC framework. The authors must present a compelling cost-benefit analysis (total token consumption vs. performance gain) to establish the economic and technical value proposition of OMAC over simpler baseline methods.

**Questions:**

1. Given that OMAC essentially seeks to optimize MAS based on task performance, how does this approach technically relate to established reinforcement learning-based methods that optimize agents and communication protocols? The authors are strongly encouraged to supplement the related work with an explicit discussion on the relative merits (e.g., sample efficiency, convergence speed, stability) of the LLM-driven meta-optimization approach versus RL approaches, and suggest a framework for quantitative comparison against key RL baselines.

2. The baseline selections are encouraged to include more advanced methods (since 2025) for comparison.

---

> ### Author Response · Authors · 2025-11-21
>
> Dear Reviewer YvM6,
>
> Thank you for your positive feedback on the significance, practical potential, and superior performance of our approach. We sincerely thank the reviewer for their thoughtful and constructive comments. We hope that our response below will clarify our methods and innovations.
>
>
> **W1. Generality**
>
> Thank you for highlighting this valuable point. We'd like to clarify that the term "Broad Optimization" in our paper emphasizes that our proposed five functional/structural dimensions for MAS optimization are general, and our proposed framework can be generally applied to optimize these dimensions, either individually or jointly. As such, our approach is intentionally general to accommodate a wide range of optimization targets for MAS.
>
> In particular, we view the optimization of an agent’s tool/API usage and internal state representation as part of the first optimization dimension (Fun-1), which is enhancing the agent’s own functional capability. In our experiments on the HumanEval benchmark, for instance, **we do optimize agents’ ability to invoke tools** to complete code generation tasks. The tools include "Syntax Checker" (for syntax validation), "Unit Tester" (equipped with a code interpreter for unit testing), etc. Agents are able to generate code with specific formats to call and use these tools, which aligns with the goal of our proposed functionality optimization.
>
> While our current practical implementation primarily focuses on prompt optimization, which may not fully address optimization under certain stricter constraints (e.g., memory management or retrieval mechanisms), we argue that these cases still conceptually belong to the category of agent functionality optimization. In that case, such optimization tasks can be naturally integrated into our framework by leveraging LLMs’ contrastive reasoning ability with positive/negative sample pairs, following our single- and multi-dimension optimization algorithms.
>
> Nevertheless, we recognize the value of this point and have included a relevant discussion in the "Future Work" section of the updated manuscript (Appendix C.2), outlining how the framework could be extended to cover broader and more complex optimization dimensions in the future.
>
>
>
> **W2. Optimization Principles**
>
>
> Thank you for this thoughtful comment. We'd like to clarify that the motivation and intuition behind our five optimization dimensions **have been detailed at Section 3.1 and Appendix A**. Specifically, these dimensions are derived by conceptualizing the multi-agent collaboration process as an **information flow through a graph**, where agents represent nodes and communication paths represent edges.
> Thus, the two functional dimensions optimize node capabilities, either improving existing agents or constructing new ones. And the three structural dimensions define graph construction, encompassing global and dynamic local agent selection, as well as inter-agent communication routes. Together, these five dimensions comprehensively capture the essential aspects required to optimize the information-flow graph of a multi-agent system.
>
> While other task-dependent considerations may arise, our proposed dimensions emphasize the core and generalizable elements (i.e., nodes and edges) that underlie multi-agent collaboration. These five dimensions are conceptually defined and introduced by us, representing a key conceptual contribution of this work.
>
> As the central theme of the paper is to establish a framework for MAS optimization, we intentionally focus on defining the optimization dimensions and methodology rather than conducting a rigorous theoretical analysis of the optimization landscape or deriving formal guarantees of optimal configurations. We'd like to leave these explorations for future work and provide a brief discussion in the newly added "Future Work" section (Appendix C.2).

---

> ### Author Response · Authors · 2025-11-21
>
> **W3. Performance and Computation Cost**
>
>
> We wish to clarify that OMAC's performance improvement should not be considered marginal. For single-dimension optimization, OMAC achieves average relative performance gains of 3.6%, 2.8%, and 4.9% per dimension across the three benchmarks, compared to the most recent and strong baselines like DyLAN (COLM 2024) [5], ADAS (ICLR 2025) [7], and AFlow (ICLR 2025) [8]. More importantly, OMAC’s ability to jointly optimize multiple dimensions yields **substantially larger improvements** than optimizing individual dimensions alone. For instance, as shown in Figure 4, joint optimization of Fun-1.1 and Fun-1.2 improves performance from 2.9% to 9.6%. Similar trends can be observed in Figures 5–6. These results demonstrate that OMAC provides systematic, meaningful, and compounding performance improvements in MAS optimization rather than incremental ones.
>
> Regarding computation cost, we wish to address that a detailed discussion of both inference and training costs is **already provided in Appendix B.2.6**. Notably, OMAC significantly reduces inference-time computational cost by dynamically optimizing the collaboration structure, outperforming other multi-agent frameworks in efficiency. This is a key advantage of OMAC, as inference efficiency is crucial for the practical deployment of agentic systems in real-world, industrial applications.
>
> For training cost, by adopting a sampling-based training strategy, OMAC can optimize a single MAS dimension for the code generation task at a cost of only approximately $5, achieving an average relative performance improvement of over 2.2% compared to the strongest baseline. And the multi-dimension optimization can provide even more significant performance improvement.
> This high cost-effectiveness makes OMAC  attractive for industrial adoption, where even performance gains of around 1% can yield substantial practical and commercial value. We believe this practical cost-performance balance underscores the technical applicability of our approach.
>
>
>
> **Q1. RL-based Methods**
>
>
> Thank you for raising this insightful point. First, we'd like to clarify that existing research applying RL-based methods to multi-step collaboration among multiple LLM-based agents remains very limited. Most prior works employing RL for agent optimization primarily target single-agent settings [1, 2], which differ fundamentally from the multi-agent optimization problem addressed in our study.
>
> While a few very recent and unpublished works, such as Spiral [3] and MHGPO [4], have taken preliminary steps toward RL-based MAS optimization, their approaches are generally restricted to single-interaction or shared-policy settings, where all agents follow the same control policy and engage in only one round of communication or problem solving. This is a substantially simpler and more constrained scenario than the one we investigate, which involves multi-step collaboration among role-specialized agents.
> Therefore, **these works are not directly comparable to ours**, particularly from an empirical or quantitative standpoint.
>
> Nevertheless, we do agree that including a conceptual comparison with these emerging studies can be beneficial by providing a broader literature context and future work potential. Following the reviewer’s suggestion, we have expanded the related work discussion in Section 5 to include these RL-based approaches and also outlined some potential directions for integrating RL methods into our framework in the "Future Work" section (Appendix C.2).
>
>
> **Q2. Baseline Selection**
>
> We'd like to address that our experiments already incorporate several recent and highly relevant baseline methods, including DyLAN (COLM 2024) [5], AgentVerse (ICLR 2024) [6], and **ADAS (ICLR 2025)** [7]. Specifically, DyLAN and ADAS are particularly recent and advanced approaches with significant citation counts in multi-agent system optimization research. Additionally, we previously included comparisons with another recent method, **AFlow (ICLR 2025)** [8], on the MBPP benchmark (Appendix B.2.5). Having allocated more resources recently, we have now extended our evaluation to all benchmarks (Table 1-3, Table 13-14), with these results included in the updated manuscript. As demonstrated both in the manuscript and the accompanying table below, our approach consistently outperforms these SOTA multi-agent system optimization methods.
>
>
> |       |ADAS |AFLOW|OMAC   |       |       |       |       |       |       |     |     |     |     |
> |-------|-----|-----|-------|-------|-------|-------|-------|-------|-------|-----|-----|-----|-----|
> |       |     |     |Fun-1.1|Fun-1.2|Fun-1.3|Fun-1.4|Fun-1.5|Fun-1.6|Fun-1.7|Fun-2|Str-1|Str-2|Str-3|
> | Pass@1|75.61|85.63|88.39  |86.31  |88.87  |89.25  |88.74  |88.39  |88.34  |86.77|86.76|86.92|87.55|

---

> > ### Author Response · Authors · 2025-11-21
> >
> > ---
> >
> >
> > References
> >
> >
> > [1]: Shao, Zhihong, et al. "Deepseekmath: Pushing the limits of mathematical reasoning in open language models, 2024." URL https://arxiv. org/abs/2402.03300 2.3 (2024): 5.
> >
> > [2]: Qian, Cheng, et al. "Toolrl: Reward is all tool learning needs." arXiv preprint arXiv:2504.13958 (2025).
> >
> > [3]: Liu, Bo, et al. "SPIRAL: Self-Play on Zero-Sum Games Incentivizes Reasoning via Multi-Agent Multi-Turn Reinforcement Learning." arXiv preprint arXiv:2506.24119 (2025).
> >
> > [4]: Chen, Guanzhong, et al. "Heterogeneous Group-Based Reinforcement Learning for LLM-based Multi-Agent Systems." arXiv preprint arXiv:2506.02718 (2025).
> >
> > [5] Liu, Zijun, et al. "A dynamic llm-powered agent network for task-oriented agent collaboration." First Conference on Language Modeling. 2024.
> >
> > [6] Chen, Weize, et al. "Agentverse: Facilitating multi-agent collaboration and exploring emergent behaviors in agents." ICLR. 2024.
> >
> > [7] Hu, Shengran, Cong Lu, and Jeff Clune. "Automated design of agentic systems." ICLR. 2025.
> >
> > [8] Zhang, Jiayi, et al. "Aflow: Automating agentic workflow generation." ICLR. 2025.

---

### Official Review · Reviewer_2GCg · 2025-10-29

**Soundness:** 3
**Presentation:** 3
**Contribution:** 2
**Rating:** 6
**Confidence:** 4

**Summary:**

The paper proposes a general, supervised optimization framework for LLM-based multi-agent systems (MAS) operating in multi-step collaborations. The authors identify five optimization dimensions—two *functional* and three *structural*. For any chosen dimension, the proposed method iterates two LLM instances: a Semantic Initializer to generate candidate prompts/controllers and a Contrastive Comparator that refines them by reasoning over supervised positive–negative pairs produced from training-set evaluations. Experiments on HumanEval (code), MMLU (general reasoning), MATH (arithmetic), and extra benchmarks show the empirical significance across single-dimension and multi-dimension settings, with ablations showing the Contrastive Comparator contributes non-trivial gains. The authors also report cost benefits at inference time via structural optimization and disclose training costs.

**Strengths:**

1. **Clear, general framework covering both agent capability and collaboration topology.** The five-dimension taxonomy is well-motivated (nodes/edges view of MAS) and seems broadly reusable.
2. **Methodological backbone is valid.** Contrastive refinement using online explored positive–negative training pairs. This approach is intuitive and straightforward to implement.
3. **Consistent improvements across tasks and dimensions.** Single-dimension results show OMAC improves over strong MAS baselines, and multi-dimension optimization yields additional gains.
4. **Rich ablations.** The effort to ablate hyperparameters, dimension contributions, and the Contrastive Comparator’s role adds confidence in the findings.

**Weaknesses:**

1. **Small supervised splits may limit statistical validity and generalization.** Using the same data points from the test sets for training raises concerns about overfitting and generalization. Though aligned with DyLAN, larger, disjoint training sets would strengthen the claims.
2. **Baseline selection could be broader.** The experiments compare against a limited set of MAS baselines. Including more recent or diverse approaches for different single-/multi-dimension optimizations would better contextualize the contributions, e.g., prompt optimization methods for agent-level prompts, or graph neural network-based MAS for structural aspects.
3. **Discussion of gradient and contextual optimization is lacking.** The proposed method relies on LLM inference calls for optimization. However, gradient-based methods are natural when contrastive pairs are available. A discussion of the trade-offs between these approaches is missing.
4. **Joint-dimension synergy is constrained.** Simultaneously optimizing multiple dimensions can hurt stability; the proposed iterative scheme is a pragmatic workaround, but deeper analysis of joint optimization remains unexplored, e.g., data mixture, gradient blending, or other multi-objective optimization techniques.

**Questions:**

How do you envision the practical training cost for MAS optimization? Though OMAC outperforms several baselines, is the training cost worth the performance gain in practical applications?

---

> ### Author Response · Authors · 2025-11-21
>
> Dear Reviewer 2GCg,
>
> Thank you for your positive feedback regarding the generality, rationale, soundness, and significance of our work. We greatly appreciate your thoughtful questions and constructive suggestions, which enable us to further improve the clarity and quality of our manuscript. We hope that our response below can be helpful to address your concerns.
>
> ---
>
> **W1. Training/Test Data Splitting**
>
> We wish to clarify that we **did not utilize any test data points during training**. As explicitly detailed in Section 3.4 and Section 4, as well as in Appendix B.1, the training and test datasets were created through completely disjoint splits, consistent with standard machine learning methodologies. Furthermore, in Appendix B.2.7, we present training curves that clearly depict performance on both the training and test sets during the optimization process. These curves demonstrate a strong correlation between training and test performance, indicating that no overfitting occurs. We kindly refer you to these sections for further details and additional discussion.
>
>
>
> **W2. Baseline Selection**
>
> We'd like to clarify that our experiments already incorporate several recent and highly relevant baseline methods, including DyLAN (COLM 2024) [1], AgentVerse (ICLR 2024) [2], and **ADAS (ICLR 2025)** [3]. Specifically, DyLAN and ADAS are particularly recent and influential approaches with significant citation counts in MAS optimization research, which focus on unsupervised prompt optimization and prompt evolution combined with search tools, respectively.
>
> Additionally, we previously included comparisons with another recent method, **AFlow (ICLR 2025)** [4], on the MBPP benchmark (Appendix B.2.5). Specifically, AFlow develops an algorithm based on Monte Carlo Tree Search for discovering effective agentic workflows.
> Having allocated more resources and time, we have now extended our evaluation to all benchmarks, with these results included in the updated manuscript (Table 1-3, Table 13-14). As clearly demonstrated both in the manuscript and the accompanying table below, our approach consistently outperforms these SOTA multi-agent system optimization methods.
>
>
>
> |       |ADAS |AFLOW|OMAC   |       |       |       |       |       |       |     |     |     |     |
> |-------|-----|-----|-------|-------|-------|-------|-------|-------|-------|-----|-----|-----|-----|
> |       |     |     |Fun-1.1|Fun-1.2|Fun-1.3|Fun-1.4|Fun-1.5|Fun-1.6|Fun-1.7|Fun-2|Str-1|Str-2|Str-3|
> | Pass@1|75.61|85.63|88.39  |86.31  |88.87  |89.25  |88.74  |88.39  |88.34  |86.77|86.76|86.92|87.55|

---

> ### Author Response · Authors · 2025-11-21
>
> **W3. Discussion of Gradient and Contextual Optimization**
>
>
> As discussed in Section 5, we acknowledge that previous research has indeed explored gradient-based optimization methods for agent systems. However, such methods **are not designed for multi-agent systems that require multi-step collaborative interactions**. In scenarios involving multiple interacting agents, gradient-based optimization becomes particularly challenging due to the presence of numerous confounding variables and the difficulty of jointly performance optimization with multiple functional and structural components. Given the primary objective of our paper, which is **leveraging LLMs' reasoning capabilities for holistic optimization** of MAS, we'd like to leave the integration of gradient-based approaches for future research.
>
> Nevertheless, we acknowledge the value of providing additional discussion on this point. Following the reviewer's suggestions, we have expanded the related work section (Section 5) to offer a more detailed overview of existing gradient-based methods. We have also added a dedicated “Future Works” section in the updated manuscript, which includes a concise yet comprehensive discussion. Please refer to Appendix C.5 for further details.
>
>
>
> **W4. Joint-Dimension Synergy**
>
> Thank you for highlighting this valuable point. As illustrated in Section 3.3, the motivation for iterative optimization across multiple dimensions originates from leveraging LLMs' contrastive reasoning capabilities. To effectively isolate and understand the sources of performance differences, **it's essential to control variations** within positive–negative pairs along a single dimension at a time, while maintaining consistency across other factors. Our experiments and discussions in Appendix B.2.3 further validate the effectiveness and necessity of this iterative approach.
>
> Since our optimization paradigm fundamentally differs from traditional gradient-based frameworks, techniques such as gradient blending, data mixing, or other multi-objective optimization methods cannot be directly applied within our setting. As our paper’s focus lies in designing an integrated framework for general MAS optimization, we prefer to leave the incorporation of other joint-dimension optimization strategies within our framework for future work. Nevertheless, we deem this as an interesting and valuable research direction and have included a relevant discussion in the "Future Works" section of the updated manuscript (Appendix C.5).
>
>
>
> **Q1. Computation Cost**
>
>
> Thank you for the question regarding the computation cost. We'd like to clarify that a detailed analysis of OMAC’s computational efficiency is **already provided in Appendix B.2.6**. As discussed there, OMAC significantly reduces inference time and computational cost by dynamically optimizing the collaboration structure, outperforming other multi-agent frameworks. We highlight this as an important advantage of OMAC, since inference efficiency is crucial for the practical deployment of agentic systems in real-world scenarios. Once training is completed, models are typically deployed for user-facing applications, where inference cost directly impacts both scalability and commercial competitiveness.
>
> In Appendix B.2.6, we also provide a detailed analysis of OMAC’s training cost. Specifically, by leveraging the sampling-based training strategy, OMAC can optimize a single MAS dimension for the code generation task with a cost of approximately only $5, achieving an average relative performance improvement of over 2.2% compared to the strongest baseline. Such efficiency makes OMAC attractive for industrial use, where even performance gains exceeding 1% can yield substantial commercial benefits. As demonstrated both in the manuscript and the accompanying table below,  we believe this practical cost-performance balance underscores the real-world applicability of our approach.
>
>
> |                   |AgentVerse |DyLAN |AFlow  |  OMAC (avg.) |
> |-------------------|-----------|------|-------|--------------|
> |Training Cost (\$) |-          |4.62  |8.24   |         5.15 |
> |Inference Cost (\$)|0.1928     |0.1288|0.1134 |        0.0974|
> |Accuracy (\%)      |78.26      |85.74 |85.63  |        87.68 |
>
>
> ---
>
> References
>
> [1] Liu, Zijun, et al. "A dynamic llm-powered agent network for task-oriented agent collaboration." First Conference on Language Modeling. 2024.
>
> [2] Chen, Weize, et al. "Agentverse: Facilitating multi-agent collaboration and exploring emergent behaviors in agents." ICLR. 2024.
>
> [3] Hu, Shengran, Cong Lu, and Jeff Clune. "Automated design of agentic systems." ICLR. 2025.
>
> [4] Zhang, Jiayi, et al. "Aflow: Automating agentic workflow generation." ICLR. 2025.

---

### Author Response · Authors · 2025-11-26

Dear Reviewers,

Thank you once again for the time and care you have dedicated to evaluating our paper and for the constructive feedback you have provided. With the reviewer–author discussion deadline (December 3) approaching **in the coming week**, we would like to confirm that we have adequately addressed all of your comments. If there are any additional questions, clarifications, or suggestions you would like us to consider, please feel free to let us know. Your insights are invaluable to us, and we are eager to address any remaining issues to further improve our work.

Best regards,

The Authors

---

### Author Response · Authors · 2025-12-02
**Rebuttal Summary**

Dear ACs and SACs,

Thank you for overseeing the review process and for all your efforts in evaluating our paper and rebuttals. We greatly appreciate your work. Below, we briefly summarize our key responses to the reviewers’ comments.

 &nbsp;
## **Baselines and Related Works**

First, we'd like to emphasize that our experiments already include several of the most recent and highly relevant MAS optimization baselines, such as DyLAN (COLM 2024) [1], AgentVerse (ICLR 2024) [2], ADAS (ICLR 2025) [2], and AFlow (ICLR 2025) [3]. Notably, ADAS represents a most recent prompt-evolution approach for MAS optimization, while AFlow is a supervised optimization method that leverages Monte Carlo Tree Search. Across all benchmarks (Tables 1–3 and 13–14), OMAC consistently outperforms these **latest 2024/2025 methods**, demonstrating the strength and generality of our framework.

Regarding related work, we clarify that existing gradient-based approaches, including SFT- and RL-based methods, remain very limited for our setting of **multi-step collaboration among multiple LLM-based agents**. Nevertheless, we have expanded our discussion of these methods and several most recent MAS optimization approaches in Section 5, and we outline additional future research directions for OMAC in Appendix C.2.



 &nbsp;

## **Cost Analysis and Performance**

We clarify that our original submission **already includes a detailed analysis of OMAC’s computational cost and performance in Appendix B.2.6**. In particular, OMAC achieves substantial improvements in both inference efficiency and overall performance compared to all baselines, while maintaining a training cost similar to the most training-efficient method.

Regarding performance improvement, we emphasize that for single-dimension optimization, OMAC achieves average relative performance gains of 3.6\%, 2.8\%, and 4.9\% across the three main benchmarks, outperforming the strongest and most recent baselines such as DyLAN, ADAS, and AFlow. More importantly, OMAC’s ability to **jointly optimize multiple dimensions leads to significantly larger improvements** than optimizing each dimension independently (e.g., increasing the gain from 2.9\% to 9.6\% in Figure 4). These results show that OMAC delivers systematic, meaningful, and compounding improvements in MAS optimization rather than incremental gains.


 &nbsp;

## **Training and Optimization Process**

First, we address that OMAC  **does not use any ground-truth labels at test time, nor does it access any test data during training**, strictly following a standard supervised learning setup as clearly illustrated in Sections 3.4 and 4.

Second, the rationale, effectiveness, and iterative design of OMAC’s optimization mechanism **have been explained in detail in Sections 3.2, 3.3, and 4.2**. Specifically, we first derive five optimization dimensions by conceptualizing multi-agent collaboration as information flow over a graph, where agents are nodes and communication paths are edges. This formulation captures the core functional and structural components essential for MAS optimization.
We then introduce two algorithms for both single-dimensional and multi-dimensional optimization. In particular, our iterative optimization paradigm for multi-dimensional settings is motivated by the need to preserve the effectiveness of the LLM’s contrastive reasoning when multiple interacting variables are involved. As shown in Appendix B.2.3, we've conducted experiments to confirm the necessity and effectiveness of this iterative strategy.

---

> ### Author Response · Authors · 2025-12-02
> **Rebuttal Summary**
>
> &nbsp;
>
> ## **Generality and Novelty**
>
> We clarify that we use the term “Broad Optimization” to highlight that the five functional and structural dimensions we propose for MAS optimization are general-purpose, and that our framework can be applied to optimize these dimensions either individually or jointly. Accordingly, OMAC is intentionally designed to accommodate a wide range of MAS optimization targets. We've demonstrated OMAC’s versatility across diverse tasks, including general reasoning, arithmetic reasoning, code generation, and challenging tool-based problem solving. These results further strengthen OMAC’s demonstrated generality to optimize advanced technical components in MAS.
>
>
> Regarding novelty, our contributions lie in the conceptual framing and integration of MAS optimization within a unified framework, as summarized in the Introduction. Specifically, we (1) identify five key optimization dimensions covering both agent functionality and collaboration structure; (2) propose a general framework using two LLM-based actors to optimize each dimension; (3) introduce an iterative algorithm for joint multi-dimension optimization; and (4) provide extensive empirical validation across multiple benchmarks, consistently outperforming strong MAS baselines.
>
> Thus, we emphasize that **the novelty does not stem from the individual optimization components** (i.e., the LLM actors), but from the conceptual generalization, unified methodology, and systematic empirical validation, which together represent a substantial and meaningful contribution to the field.
>
>
>  &nbsp;
>
> ## **Additional Experiments**
>
>
> Following the reviewers’ suggestions, we've conducted several additional experiments, such as evaluations using two open-source backbone LLMs, added results on the recent and challenging GAIA benchmark, and a more direct and comprehensive cost analysis. Across all these expanded experimental settings, OMAC consistently outperforms all baselines, further demonstrating its strong generalization ability, effectiveness, and efficiency. We’d like to emphasize that these additional experiments have **substantially addressed reviewer KmRj’s concerns, leading him to raise his score** (as the only reviewer responding to our rebuttal).
>
>
>
>
> ---
>
> [1] Liu, Zijun, et al. "A dynamic llm-powered agent network for task-oriented agent collaboration." First Conference on Language Modeling. 2024.
>
> [2] Chen, Weize, et al. "Agentverse: Facilitating multi-agent collaboration and exploring emergent behaviors in agents." ICLR. 2024.
>
> [3] Hu, Shengran, Cong Lu, and Jeff Clune. "Automated design of agentic systems." ICLR. 2025.
>
> [4] Zhang, Jiayi, et al. "Aflow: Automating agentic workflow generation." ICLR. 2025.

---

### Meta-Review · Area_Chair_8iR8 · 2026-01-08

**Summary:**

This paper proposes to optimize LLM-based multi-agent systems by decomposing MAS design into five dimensions, and iteratively refining prompts and collaboration structures using contrastive reasoning. Reviewers broadly agree that the paper is clearly written and presents a systematic and reusable framework. However, Reviewers also have shared concerns that the present solution is incremental to existing prompt evolution, rejection fine-tuning, and agent optimization methods. Besides, Reviewers also point out insufficient baseline coverage and incomplete cost–benefit analysis. While the rebuttal clarifies several points and adds experiments, these improvements are insufficient to overcome the fundamental concerns about novelty and empirical rigor. Therefore, I recommend rejection.

**Reviewer Concerns:**

Adding comparisons to newer baselines;
Expanding cost analysis;
Including additional benchmarks and open-source models.

**Reviewer Scores:**

Reviewer KmRj would raise the rating to 4.

---

### Decision · Program_Chairs · 2026-01-26

Reject